

# When does vapor pressure deficit drive or reduce evapotranspiration?

Adam Massmann[1], Pierre Gentine[1], and Changjie Lin[1,2]

[1]Department of Earth and Environmental Engineering, Columbia University, New York, NY 10027
[2]State Key Laboratory of Hydroscience and Engineering, Department of Hydraulic Engineering, Tsinghua University, Beijing, CN 100084

**Correspondence:** Adam Massmann (akm2203@columbia.edu)

**Abstract.** Increasing vapor pressure deficit (VPD) increases atmospheric demand for water, and vapor pressure deficit is expected to rise with increasing greenhouse gases. While increased evapotranspiration (ET) in response to increased atmospheric demand seems intuitive, plants are capable of reducing ET in response to increased VPD by closing their stomata, in an effort to conserve water. Here we examine which effect dominates response to increasing VPD: atmospheric demand and increases in ET, or plant physiological response (stomata closure) and decreases in ET. We use Penman-Monteith, combined with semi-empirical optimal stomatal regulation theory and underlying water use efficiency, to develop a theoretical framework for understanding how ET responds to increases in VPD.

The theory suggests that for most environmental conditions and plant types, plant physiological response dominates and ET decreases with increasing VPD. Plants that are evolved or bred to prioritize primary production over water conservation (e.g. crops) exhibit a higher likelihood of atmospheric demand-driven response (ET increasing). However for forest, grass, savannah, and shrub plant types, ET more frequently decreases than increases with rising VPD. This work serves as an example of the utility of our simplified framework for disentangling land-atmosphere feedbacks, including the characterization of ET response in an atmospherically drier, enriched $CO_2$ world.

## 1 Introduction

Vapor pressure deficit (VPD) is expected to rise over continents in the future due to the combination of increased temperature and, depending on region, decreased relative humidity (Byrne and O'Gorman, 2013). Increases in VPD increase the atmospheric demand for evapotranspired water (Penman, 1948; Monteith et al., 1965), but also stress plant stomata (Leuning, 1990; Medlyn et al., 2011).

The opposing effects of increased atmospheric demand and higher stomatal stress lead to two possible perspectives for how evapotranspiration (ET) responds to shifts in VPD. The first, a hydrometeorological perspective, is that higher VPD increases atmospheric demand for water from the land surface, and this drives an increase in evapotranspiration (ET). This perspective is particularly relevant because potential evapotranspiration (PET), which is used in many drought indices and hydrometeorological studies (e.g., Heim Jr, 2002; Scheff and Frierson, 2015), typically only quantifies changes in atmospheric demand and fails to account for ecosystem response (Swann et al., 2016). In reality, plants' stomata have evolved to optimally





regulate the exchange of water and carbon, and tend to partially close in response to increased atmospheric dryness (Farquhar, 1978; Ball et al., 1987; Leuning, 1990; Medlyn et al., 2011). This leads to a plant physiology perspective, in which an increase in VPD, particularly in well-watered soil conditions, may actually correspond to a decrease in ET because of stomatal closure (e.g. Rigden and Salvucci, 2017). In other words, the question "When does VPD drive or reduce ET?" can be related to whether

plant regulation or atmospheric demand dominates ET response.

The ET response to changes in VPD alters water partitioning between the soil and atmosphere. If ecosystem plant response reduces ET with atmospheric drying then soil moisture will be better conserved. This represents a sensible evolutionary strategy to cope with aridity: save water for periods when atmospheric demand for water is relatively low, and atmospheric carbon can be accessed with a relatively smaller cost in water loss. If instead stomata were fully passive (similar to soil pores, e.g. Or

et al., 2013), increased atmospheric aridity would strongly reduce soil moisture (Berg et al., 2017). This could further increase aridity as low soil moisture levels increases the Bowen ratio, leading to increased temperature and atmospheric drying (Bouchet, 1963; Morton, 1965; Brutsaert, 1999; Ozdogan et al., 2006; Salvucci and Gentine, 2013; Gentine et al., 2016; Berg et al., 2016). Therefore, passive regulation and a lack of soil moisture conservation does not seem to be a sensible strategy for plants from an evolutionary standpoint.

As a counterpoint, one may argue that increases in ET with increasing VPD could increase the likelihood of precipitation (e.g., Findell et al., 2011). However, increases in ET do not always guarantee an increased likelihood of precipitation, which depending on environmental conditions could cause a decrease in the likelihood of precipitation (Gentine et al., 2013). Furthermore, any increases in precipitation are likely to be non-local, such that plants giving up water to the atmosphere are not guaranteed to reap the benefits of water returned from the atmosphere to the soil. Using this subjective logic, from an ecosystem

evolutionary perspective, water stored in soil seems to be worth much more than the chance of water returned as precipitation.

We can use intuition about plant water conservation strategy to hypothesize about ET response to changes in VPD. Plants and ecosystems that evolved to conserve water, such as arid shrubs or savannah, should be more likely to reduce ET with increasing VPD, and plants that have evolved or have been engineered to care little about water, such as crops, will be more likely to increase ET with increasing VPD. Atmospheric conditions must matter as well. At the ecosystem scale, there are

limits to plant water conservation strategies. As atmospheric demand for water (VPD) increases, ecosystems should begin to reach their water conservation limits and might not be able to entirely limit ET flux to the atmosphere. At this stage any further increase in VPD will most likely drive a (limited) increase in ET, because the increase in atmospheric demand for water overwhelms the limited plant response to conserve water.

The objective of the present manuscript is to use reasonable approximations as a tool to develop intuition for plant response

to atmospheric drying and evaluate the VPD dependence of ET. This intuition will aid interpretation of observations, full complexity models, and facilitate the disentanglement of complex land-atmosphere feedbacks. In the past, similar approaches were used to understand interactions between stomatal conductance, evapotranspiration and the environment (e.g., Jarvis and McNaughton, 1986; McNaughton and Jarvis, 1991). However, at the time researchers' understanding of the form of VPD's effect on plant physiology was limited, so they could not explore the sensitivity of ET to VPD, including VPD's effect on

stomatal conductance and plant function.



Recent results have drastically improved our understanding of VPD's impact on physiology, especially at the leaf level. Medlyn et al. (2011) developed a model for leaf-scale stomatal conductance ($g_s$), including VPD response, by combining an optimal photosynthesis theory (Cowan and Farquhar, 1977) with an empirical approach, and extended use of this model to the ecosystem scale in Medlyn et al. (2017). Additionally, Zhou et al. (2014) demonstrated that a quantity underlying water use efficiency $\left( uWUE = \frac{GPP \sqrt{VPD}}{ET} \right)$ properly captures a constant relationship between GPP, ET, and VPD over a diurnal cycle at the ecosystem scale. uWUE is also remarkably well conserved in the growing season across space and time, within a PFT (Zhou et al., 2015). While stomatal conductance parameterizations and uWUE greatly simplify complex plant physiological processes, they still capture leading order ecosystem behavior for vegetated surfaces (Medlyn et al., 2017; Zhou et al., 2014), and are novel tools to transparently develop intuition for the behavior of complex, multiscale ecohydrologic systems.

In this manuscript, we leverage uWUE and recent developments in stomatal conductance parameterizations (Medlyn et al., 2011) to derive the theoretical one-way response of ET to VPD with other environmental variables properly controlled for, i.e. we develop a framework for evaluating the partial derivative of ET with respect to VPD. For the first time, we explicitly include VPD's full effect on stomatal conductance, including its leading order impact on photosynthesis. Our theory is validated and tested at multiple eddy-covariance stations spanning various climates and plant functional types.

## 2 Materials and Methods

### 2.1 Data

We use both meteorological and eddy-covariance data from the FLUXNET2015 database (data available at https://fluxnet.fluxdata.org/data/fluxnet2015-dataset/ ), including all Tier 1 sites with at least four years of data and observations of the variables described in the methods section (Sect. 2.2). Sixty-six sites met these requirements, and were grouped into nine plant functional types (PFT) according to the International Geosphere-Biosphere Programme vegetation classification scheme (Loveland et al., 1999): cropland (CRO), grass (GRA), deciduous broadleaf forest (DBF), evergreen broadleaf forest (EBF), evergreen needleleaf forest (ENF), mixed forest (MF), closed shrub (CSH), savannah (SAV), and woody savannah (WSA) (site locations and citations are in Fig. 1 and Appendix A, respectively).

The purpose of this study is to examine ecosystem response to atmospheric drying, focusing on the growing season. To accomplish this, we filter and quality control the data using a similar procedure as Zhou et al. (2015):

- Only measured or highest ("good") quality gapfilled data, according to quality control flags, are used.

- To isolate the growing season, we only use days in which the average Gross Primary Productivity (GPP) exceeds 10% of the observed 95th percentile of GPP for a given site. GPP is calculated using the nighttime respiration partitioning method.

- We remove days with rain and the day following to avoid issues with rain interception and sensor saturation at high relative humidity (Medlyn et al. (2017)).





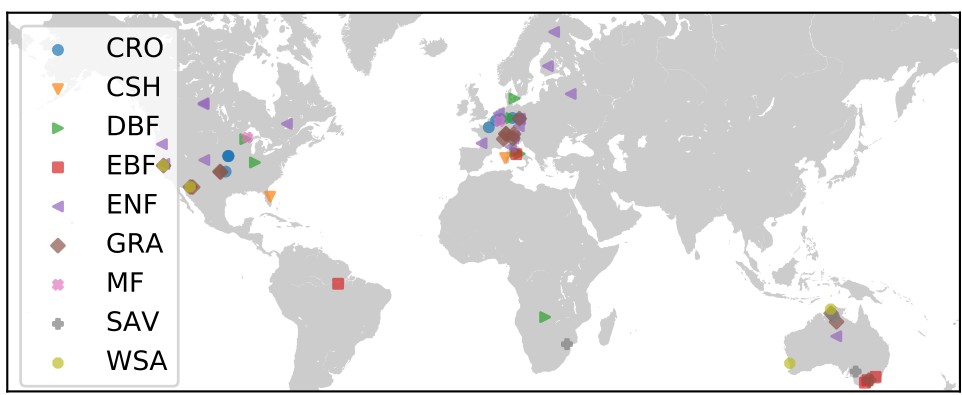

**Figure 1.** Plant functional type and location of FLUXNET2015 sites used in this analysis.

Additionally, as in Lin et al. (2018), we restrict data to the daytime, which is identified when downwelling shortwave radiation is greater than 50 W m$^{-2}$ and sensible heat flux is greater than 5 W m$^{-2}$. To reduce the chance of sensor saturation at high relative humidity, we remove all time steps for which VPD is less than .01 kPa, and to reduce errors at low windspeeds we remove all periods with wind magnitudes less than 0.5 m s$^{-1}$. Timesteps with negative observed GPP or ET are also removed, and we aggregate half hourly data to hourly averages to reduce noise (Lin et al., 2018). After these quality control procedures, 400,983 upscaled hourly observations remain.

## 2.2 Methods

The Penman-Monteith equation (hereafter PM, Penman, 1948; Monteith et al., 1965) estimates ET as a function of observable atmospheric variables and surface conductances:

$$ET = \frac{\Delta R_{net} + g_a \rho_a c_p VPD}{\Delta + \gamma(1 + \frac{g_a}{g_s})},$$ (1)

where $\Delta$ is the change in saturation vapor pressure with temperature, given by Clausius-Clapeyron ($\frac{d\,e_s}{d\,T}$), $R_{net}$ is the net radiation minus ground heat flux, $g_a$ is aerodynamic conductance, $\rho_a$ is air density, $c_p$ is specific heat of air at constant pressure, $\gamma$ is the psychometric constant, and $g_s$ is the stomatal conductance (Table 1).

Medlyn et al. (2011) developed a model for stomatal conductance ($g_s$) by combining an optimal photosynthesis theory (Cowan and Farquhar, 1977) with an empirical approach, which describes the dependence of $g_s$ to VPD. This resulted in the following model for leaf-scale stomatal conductance:

$$g_{s-leaf} = g_0 + 1.6\left(1 + \frac{g_{1-leaf}}{\sqrt{VPD}}\right)\frac{A}{c_a},$$ (2)




**Table 1.** Definition of symbols and variables, with citation for how values are calculated, if applicable.

| Variable | Description | Units | Citation |
|---|---|---|---|
| $e_s$ | saturation vapor pressure | Pa | - |
| $T$ | temperature | K | - |
| $P$ | pressure | Pa | - |
| $\Delta$ | $\frac{\partial e_s}{\partial T}$ | Pa K$^{-1}$ | - |
| $R_{net}$ | net radiation at land surface minus ground heat flux | W m$^{-2}$ | - |
| $g_a$ | aerodynamic conductance | m s$^{-1}$ | Shuttleworth (2012) |
| $\rho_a$ | air density | kg m$^{-3}$ | - |
| $c_p$ | specific heat capacity of air at constant pressure | J K$^{-1}$ kg$^{-1}$ | - |
| $VPD$ | vapor pressure deficit | Pa | - |
| $\gamma$ | psychometric constant | Pa K$^{-1}$ | - |
| $g_{s-leaf}$ | leaf-scale stomatal conductance | m s$^{-1}$ | Medlyn et al. (2011) |
| $g_s$ | stomatal conductance | m s$^{-1}$ | Medlyn et al. (2017) |
| $g_{1-leaf}$ | leaf-scale slope parameter | Pa$^{0.5}$ | Medlyn et al. (2011) |
| $g_1$ | ecosystem-scale slope parameter | Pa$^{0.5}$ | Medlyn et al. (2017) |
| $R$ | universal gas constant | J mol$^{-1}$ K$^{-1}$ | - |
| $R_{air}$ | gas constant of air | J K$^{-1}$ kg$^{-1}$ | - |
| $\sigma$ | uncertainty parameter | - | - |
| $c_a$ | CO$_2$ concentration | $\mu$ mol CO$_2$ mol$^{-1}$ air | - |
| $\lambda$ | marginal water cost of leaf carbon | mol H$_2$O mol$^{-1}$ CO$_2$ | - |
| $\Gamma$ | CO$_2$ compensation point | - | - |
| $\Gamma^*$ | CO$_2$ compensation point without dark respiration | - | - |

where $g_{1-leaf}$ is a leaf-scale "slope" parameter, A is the net CO$_2$ assimilation rate, and $c_a$ is the atmospheric CO$_2$ concentration at the leaf surface. Medlyn et al. (2011) relate the slope parameter ($g_{1-leaf}$) to physical parameters as:

$$g_{1-leaf} = \sqrt{\frac{3\Gamma^* \lambda}{1.6}},^1 \tag{3}$$

where $\Gamma^*$ is the CO$_2$ compensation point for photosynthesis (without dark respiration), and $\lambda$ is the marginal water cost

5 of leaf carbon ($\frac{\partial \text{ transpiration}}{\partial A}$). So, $g_{1-leaf}$ is a leaf-scale term reflecting the trade-off of water for carbon uptake. The higher $g_{1-leaf}$, the more open the stomata and the more they release water in exchange for carbon.

The Medlyn model for stomatal conductance has been shown to behave very well across PFTs (Lin et al., 2015), and has been successfully adopted to ecosystem scale analysis in Medlyn et al. (2017). In units of m s$^{-1}$, the ecosystem scale stomatal conductance is:

$$10 \quad g_s = \frac{RT}{P} 1.6 \left(1 + \frac{g_1}{\sqrt{VPD}}\right) \frac{GPP}{c_a}, \tag{4}$$





**Table 2.** Plant functional types, their abbreviation, calculated Medlyn coefficient, and calculated uWUE. uWUE from Zhou et al. (2015), including the observed standard deviation, is shown for comparison. Note that uWUE from Zhou et al. (2015) is calculated from a different set of sites, and that units are converted such that the quantities work with Equations 1-8 and the variables defined Table 1.

| Abbreviation | PFT | $g_1$ (Pa$^{0.5}$) | uWUE ($\mu$-mol [C] Pa$^{0.5}$ J$^{-1}$ [ET]) | |
| --- | --- | --- | --- | --- |
| | | | fitted | Zhou et al. (2015) |
| CRO | Crops | 140.67 | 2.85 | $3.80 \pm 1.01$ |
| DBF | Deciduous Forest | 117.26 | 2.96 | $3.12 \pm 0.52$ |
| EBF | Evergreen Broadleaf Forest | 101.92 | 3.12 | N/A |
| SAV | Savannah | 96.07 | 2.79 | N/A |
| GRA | Grass | 145.56 | 2.13 | $2.68 \pm 0.61$ |
| MF | Mixed Forest | 79.23 | 3.68 | $2.99 \pm 0.62$ |
| WSA | Woody Savannah | 117.36 | 2.22 | $2.88 \pm 0.38$ |
| ENF | Evergreen Needleleaf Forest | 100.54 | 2.73 | $3.30 \pm 0.91$ |
| CSH | Closed Shrub | 75.08 | 2.82 | $2.18 \pm 0.44$ |

where GPP is the ecosystem scale gross primary production, and $g_1$ is an ecosystem scale analogue to $g_{1-leaf}$. We solve for $g_1$ following Medlyn et al. (2017) (Eq. (5)), and take the median $g_1$ value to be representative of each PFT (Table 2) instead of the mean to avoid extra weighting of rare outliers.

While Eq. (3) can be used in PM (Eq. (1)), it will make analytical work with the function intractable because $GPP$ is functionally related to ET itself. Additionally, a perturbation to VPD should induce a physiological plant response that will alter GPP and cause an indirect change in stomatal conductance, in addition to the direct effect of VPD in Eq. (4). Therefore, in order to derive the response of ET to VPD, we must account for the functional relationship between GPP, ET, and VPD, and its effect on stomatal conductance. We can use aforementioned semi-empirical results of Zhou et al. (2015) as a tool to approach this problem. Zhou et al. (2015), showed that underlying Water Use Efficiency (uWUE):

$$uWUE = \frac{GPP \cdot \sqrt{VPD}}{ET} \tag{5}$$

is relatively constant across time and moisture conditions within a plant functional type, and correctly captures a constant relationship between GPP, ET and VPD over a diurnal cycle (Zhou et al., 2014). The theoretical derivation of the square root VPD dependence in $uWUE$ leverages the same assumptions used in Medlyn et al. (2011) to derive the square-root VPD dependence of the stomatal conductance model (Eq. (4)). We can use uWUE to remove the $GPP$ dependence from $g_s$ in a way that makes PM analytically tractable:

$$g_s = \frac{RT}{P} 1.6 \left(1 + \frac{g_1}{\sqrt{VPD}}\right) \frac{uWUE \, ET}{c_a \, \sqrt{VPD}}. \tag{6}$$





Plugging Eq. (6) into Eq. (1) and rearranging gives a new explicit expression for PM, in which dependence on $GPP$ is removed:

$$ET = \frac{\Delta R_{net} + \frac{g_a\,P}{T}\left(\frac{c_p VPD}{R_{air}} - \frac{\gamma c_a \sqrt{VPD}}{R\,1.6\;\text{uWUE}\,(1+\frac{g_1}{\sqrt{VPD}})}\right)}{\Delta + \gamma} \tag{7}$$

By accounting for photosynthesis changes in ecosystem conductance, with Eq. (7) we have derived for the first time, using recent results (Medlyn et al., 2011; Zhou et al., 2014, 2015; Medlyn et al., 2017), ET explicitly as function of environmental variables and two plant-specific constants, the slope parameter ($g_1$), and uWUE, both reflecting water conservation strategy. The slope parameter is related to the willingness of stomata to trade water for $CO_2$ and to keep stomata open. uWUE is a semi-empirical ecosystem-scale constant related to how WUE changes with VPD (specifically $VPD^{-1/2}$). It is also roughly proportional to physical constants:

$$uWUE \overset{\propto}{\sim} \sqrt{\frac{c_a - \Gamma}{1.6\lambda}},$$

where $\Gamma$ is the $CO_2$ compensation point (Eq. (5) in Zhou et al., 2014). So uWUE is related to atmospheric $CO_2$ concentration and compensation point, and is inversely proportional to the marginal water cost of leaf carbon.

Given eddy-covariance FLUXNET2015 data (Sect. 2.1), every term in our new version of PM (Eq. (7)) is observed, except for uWUE, which we fit by calculating its expectation, given the model and FLUXNET2015 data.

However, eddy-covariance data are inherently noisy so we include a measure of uncertainty in our analysis. To account for observational error, as well as model uncertainty (e.g. temporal and spatial variations of uWUE and $g_1$), we introduce an uncertainty parameter $\sigma$ modifying uWUE:

$$ET = \frac{\Delta R_{net} + \frac{g_a\,P}{T}\left(\frac{c_p VPD}{R_{air}} - \frac{\gamma c_a \sqrt{VPD}}{R\,1.6\,\sigma\;\text{uWUE}\,(1+\frac{g_1}{\sqrt{VPD}})}\right)}{\Delta + \gamma} \tag{8}$$

Now, from each FLUXNET2015 observation (i.e. for each hourly observation at every time step) we can evaluate $\sigma$:

$$\sigma = -\frac{g_a \gamma c_a \sqrt{VPD} L_v P}{\left(\text{ET}\,(\Delta+\gamma) - \Delta R_{net} - g_a \rho_a c_p VPD\right)1.6\,R\,T\;\text{uWUE}\,(1+\frac{g_1}{\sqrt{VPD}})}, \tag{9}$$

So, with this uncertainty analysis we can evaluate departure from our theory in observations, as a departure of $\sigma$ from unity. The variability of $\sigma$ across sites and time provides a measure of uncertainty in our model, assumptions, as well as in the FLUXNET2015 observations themselves. The variability of $\sigma$ then propagates any uncertainty through to our partial derivative of Eq. (8) with respect to VPD:

$$\frac{\partial\,ET}{\partial\,VPD} = \frac{2\,g_a\,P}{T(\Delta+\gamma)}\left(\frac{c_p}{R_{air}} - \frac{\gamma c_a}{1.6\,R\,\sigma\;\text{uWUE}}\left(\frac{2g_1 + \sqrt{VPD}}{2(g_1+\sqrt{VPD})^2}\right)\right) \tag{10}$$





With Eq. (10) we have an analytical framework for ecosystem response to atmospheric demand perturbations with environmental conditions held fixed. There are a few subtleties to taking the derivative in Eq. (10): $\Delta\left(\frac{de_s}{dT}\right)$ and $VPD$ are functionally related, so while taking the derivative we evaluate $\frac{\partial ET}{\partial VPD} = \frac{\partial ET}{\partial e_s}\frac{\partial e_s}{\partial VPD}\Big|_{\text{RH fixed}} + \frac{\partial ET}{\partial RH}\frac{\partial RH}{\partial VPD}\Big|_{e_s \text{ fixed}}$. $RH$ and $e_s$ are assumed to be approximately independent, which is supported by the data (not shown).

This derivation relied either implicitly or explicitly on several assumptions. First, we assume that VPD at the leaf surface is the same as VPD at measurement height; physically this implies that leaves are perfectly coupled to the atmosphere. In reality, for some conditions and plant types the leaves can become decoupled from the boundary layer (De Kauwe et al., 2017; Medlyn et al., 2017). Given our focus on the growing season, which is usually characterized by relatively high insolation inducing instability and convective boundary layers, we would expect the surface to be generally well coupled. An additional

assumption in the formulation of $uWUE$ (Zhou et al., 2014, 2015) and Medlyn et al. (2017)'s stomatal conductance model is that direct soil evaporation (E) contributions to ET remain small relative to transpiration (T). This should be more true during the growing season. The ratio of E to T may increase immediately after rainfall events due to high soil moisture and ponding, but VPD is generally low anyways during these times. However, some plant types allow for systematically larger contributions of E in ET, particularly those with sparse canopies and smaller relative amounts of transpiration. We therefore might expect

that the theory will be most applicable to forest PFTs, which will be most strongly coupled to the boundary layer due to larger surface roughness, and will also generally have the highest ratios of transpiration to evaporation. Finally, for the goal of developing PFT-wide intuition, we assume that $g_1$ and $uWUE$ are constant in space and time. Both quantities have been shown to be well conserved over space and time within a PFT relative to inter-PFT variability. However, we would expect them to vary somewhat with environmental conditions including very low soil water content, temperature, and with intra-PFT

plant-specific characteristics like wood density for tree PFTs (Lin et al., 2015). Given these strong assumptions made with the goal of understanding broad, leading order plant behaviors, we take the extra care of including and analyzing uncertainty (Sect. 3.4) and examining sensitivity to the exponent of the VPD response (Sect. 3.7).

    We note one final comment on our derivation which is relevant for drought indices. If we approximate $c_a$ at a global mean $CO_2$ concentration, then the RHS of Eq. (7) is fully defined using commonly available weather station data and the constants

published here. This makes Eq. (7) useful in addition to PET in drought indices and hydrometeorological analysis for vegetated surfaces. Equation (7) better reflects the physics of water exchange at the land surface and would only require fitting of uWUE and $g_1$, and can also account for changes in $CO_2$ concentration (assuming some relationship between $\lambda$ and $[CO_2]$), which is missing in other drought indices such as the Palmer Drought Severity Index (PDSI) (Swann et al., 2016; Lemordant et al., 2016, 2018). Lastly, all code and data used in this analysis, including those used to generate the figures and tables, are publicly

available at https://github.com/massma/climate_et.

## 3 Results and Discussion

By construction, the variability in the $\sigma$ term (Eq. (9)) contains all model and observational uncertainties. For an observation that perfectly matches our model and constant uWUE assumption, $\sigma$ will be one. Therefore, for our assumptions and framework




to be reasonable $\sigma$ should be close to 1. An additional concern is that $\sigma$ may in fact be correlated with $VPD$, in which case the dependence would need to be accounted for when taking the derivative. Fortunately, there is a very weak dependence of $\sigma$ on VPD in their joint distribution, and $\sigma$ is indeed close to unity i.e. $O(1)$ (Fig. 2). Given this weak dependence and the distribution of $\sigma$ we have confidence in our model framework and the data quality.

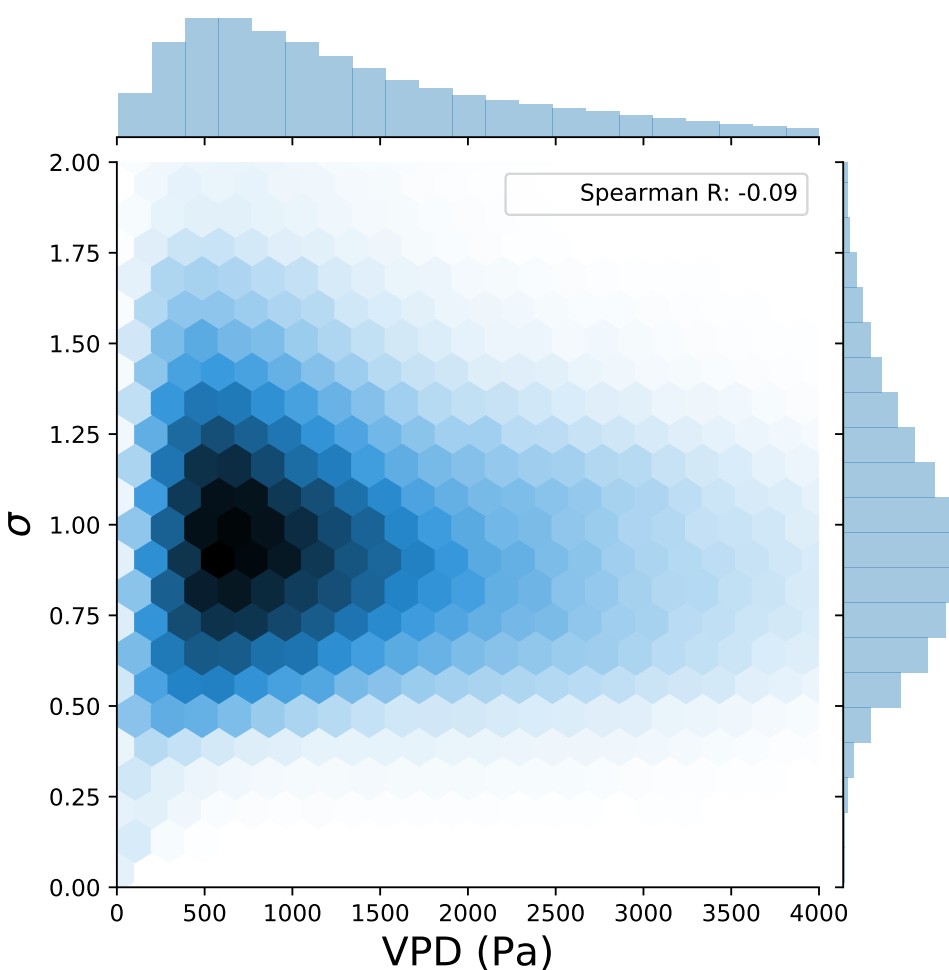

**Figure 2.** The joint distribution of $VPD$ and $\sigma$, with outliers removed (defined as lowest and highest 5% of $\sigma$). $\sigma$ exhibits a weak dependence on $VPD$, and $\sigma$ is $O(1)$ for the bulk of the observations.

5      Before calculating the sensitivity of ET to VPD, we will consider the functional form of Eq. (10). There are two main terms: a "scaling" term, which modifies the magnitude but not the sign of the ET response to VPD ($\frac{\partial ET}{\partial VPD}$):

$$\frac{g_a \, P}{T(\Delta + \gamma)},$$

(11)



and a "sign" term, which determines whether ET increases or decreases with VPD (i.e. atmospheric demand driven or physiologically controlled):

$$\frac{c_p}{R_{air}} - \frac{\gamma c_a}{1.6\, R\; \text{uWUE}} \left( \frac{2g_1 + \sqrt{VPD}}{2(g_1 + \sqrt{VPD})^2} \right). \tag{12}$$

All variables are positive, so the relative magnitude between the first term and the second term in the sign term (Eq. (12)) will determine whether ET increases or decreases with increasing VPD. If the second term is larger then plant control dominates and ET decreases with increasing VPD. However, if the first term is larger, then atmospheric demand dominates and ET increases with increasing VPD.

### 3.1 Functional Form of the Sign Term

First, we explore the variables within the sign term to gain better intuition on the driver of either the increase or reduction of ET with VPD. $CO_2$ concentration ($c_a$) and the psychometric constant ($\gamma$) are relatively constant over the dataset considered here so that the variability is dominated by $\sigma$ and $VPD$. uWUE could vary with soil moisture but has been shown to be relatively constant (Zhou et al., 2015). This then means that the sign term only depends on VPD for a given PFT and is approximately just a function of $VPD$. We can further determine a critical threshold separating an increase from a decrease in ET, i.e. the threshold $VPD_{crit}$ such that the derivative vanishes $\frac{\partial ET}{\partial VPD} = 0$:

$$VPD_{crit} = \frac{R_{air}}{4c_p} \left( \frac{\gamma c_a}{1.6\, R\, uWUE} + \sqrt{\frac{\gamma c_a}{1.6\, R\, uWUE} \left( \frac{\gamma c_a}{1.6\, R\, uWUE} + 8g_1 \frac{c_p}{R_{air}} \right)} - 4g_1 \frac{c_p}{R_{air}} \right), \tag{13}$$

noting that $VPD_{crit}$ mostly depends on the PFT parameters uWUE and $g_1$, and only varies weakly with climate as most other parameters related to the environment are nearly constant. The calculated value of $VPD_{crit}$ for each PFT is shown in Table 3. For any values of $VPD$ less than $VPD_{crit}$, ET will decrease with increasing VPD ($\frac{\partial ET}{\partial VPD} < 0$), and for values of $VPD$ greater than $VPD_{crit}$, ET will increase with increasing VPD ($\frac{\partial ET}{\partial VPD} > 0$). In other words, ecosystems regulate and mitigate evaporative losses up to the VPD limit, $VPD_{crit}$, above which atmospheric demand is just too high to be entirely compensated by stomatal and ecosystem regulation. We note however that even though ET increases again above the critical threshold, $VPD_{crit}$, ET is still much lower than potential evaporation as stomata are still strongly regulating vapor fluxes to the atmosphere. However, even in the absence of soil pore evaporation ET cannot go completely to zero at high VPD, because stomata are still slightly open to perform some photosynthesis (Ball et al., 1987; Leuning, 1990; Medlyn et al., 2011). In addition, upward xylem transport is necessary to maintain phloem transport, as well as nutrient transport and thus carbon allocation (De Schepper et al., 2013; Nikinmaa et al., 2013; Ryan and Asao, 2014).

Differences in $VPD_{crit}$ are exclusively determined by uWUE and the slope parameter ($g_1$) related to the plant functional type. A larger uWUE means a smaller $VPD_{crit}$, and an ET response to increases VPD that is more likely to be positive. At first glance this result is somewhat counter-intuitive; we expect that plants with a higher water use efficiency would be more water conservative. However, in reality uWUE determines how $WUE$ changes with VPD:





**Table 3.** Values of $VPD_{crit}$, where $\frac{\partial ET}{\partial VPD} = 0$, evaluated at PFT average values for $R_{air}$, $\gamma$, and $c_a$. PFT-specific constants ($g_1$, uWUE) are provided in Table 2. For values of $VPD$ less than $VPD_{crit}$, $\frac{\partial ET}{\partial VPD}$ will be negative, and for values of $VPD$ greater than $VPD_{crit}$, $\frac{\partial ET}{\partial VPD}$ will be positive.

| PFT | $R_{air}$ | $c_a$ (ppm) | $\gamma$ | $VPD_{crit}$ **(Pa)** |
|-----|-----------|-------------|----------|------------------------|
| CRO | 288.6 | 376.1 | 65.2 | **812.8** |
| DBF | 288.7 | 379.5 | 63.3 | **1300.0** |
| EBF | 288.3 | 366.4 | 61.5 | **1130.9** |
| SAV | 288.8 | 374.2 | 66.6 | **3502.6** |
| GRA | 288.4 | 379.1 | 60.6 | **2943.6** |
| MF | 288.2 | 384.1 | 63.5 | **1530.1** |
| WSA | 288.4 | 376.0 | 64.6 | **5234.9** |
| ENF | 288.1 | 379.2 | 60.5 | **2443.6** |
| CSH | 289.0 | 383.6 | 67.5 | **5399.0** |

$$WUE = \frac{GPP}{ET} = \frac{uWUE}{\sqrt{VPD}}.$$

$$\frac{\partial WUE}{\partial VPD} = -\frac{uWUE}{2\,VPD^{3/2}}$$

So, plants with a higher uWUE will have a greater *decrease* in ecosystem-scale $WUE$ in response to increases in VPD. This

decrease in $WUE$ causes more water loss per unit carbon gain, and explains the relationship between high uWUE and high likelihood of increases of ET in response to increasing atmospheric drying (increases in VPD).

A tendency towards increasing ET response with increasing VPD can also be caused by a high slope parameter ($g_1$), characteristic of plants that at the leaf scale are more willing to trade water for access to atmospheric $CO_2$. Plants that are less conservative will be thus be more likely to increase ET with increasing VPD. Both the aforementioned effects (large uWUE,

$g_1$) can amplify each other, and generally conspire to shift the sign term towards a positive value for a given PFT.

This effect of uWUE and $g1$ on the sign term is most apparent by comparing two extreme PFTs: water intensive crops (CRO) and water conservative closed shrub (CSH). CRO has higher slope parameter and a slightly higher uWUE ($g_1 = 140.7$ $Pa^{1/2}$; 2.85 $\mu$-mol [C] $Pa^{0.5}$ $J^{-1}$ [ET]) compared to CSH ($g1 = 75.1\,Pa^{1/2}$, $uWUE = 2.82$ $\mu$-mol [C] $Pa^{0.5}$ $J^{-1}$ [ET]). These differences in PFT parameters cause opposite ET responses to changes in VPD between CRO and CSH. ET theoretically

always decreases with increasing VPD for the more water conservative CSH, while ET frequently increases with increasing VPD for the more water intensive CRO (Fig. 3). CROs evolved or were bred to prioritize GPP and yield and are thus not water conservative. They are very willing to trade water for photosynthesis and productivity, despite changes in VPD, while CSH are very unwilling to trade water for more photosynthesis.

As expected, the slope parameter ($g_1$) is a primary determinant of the VPD dependence for the sign term shown in Fig.

3. Plants that are more conservative (small $g_1$) will tend to reduce ET with increasing VPD, and will be very effective at





**Figure 3.** The functional form of the sign term, with $\sigma$ held fixed at 1, and all terms except VPD set to PFT averages. For comparison, the observed range of VPD for each PFT is plotted below the x-axis. Stars denote 25th, 50th, and 75th percentiles, and the range of the line spans the 5th-95th percentiles of observed VPD. Vertical black lines denote the location of $VPD_{crit}$ for each PFT, with the exception of CSH and WSA, for which $VPD_{crit}$ is off-scale.



reducing ET, especially at low VPD. However, at very high VPD, gradients in vapor pressure at the leaf scale will become very strong as stomata reach their limits of closure in response to VPD (parameterized with $g_1$). As a result, ET response will begin to asymptote towards a constant ecosystem-scale values as leaf-scale response to VPD asymptotes towards zero. Therefore, plants with a low $g_1$ will have the largest VPD dependence of ET response because the difference in ET response at low VPD

(leaf stomatal response dominates) and high VPD (VPD gradient dominates) is largest. This is apparent in the strong VPD dependence of CSH, which has the lowest slope parameter ($g_1 = 75.1 \, \mathrm{Pa}^{1/2}$) (Fig. 3).

To summarize our theoretical insights (Fig. 3 and Table 3), CROs are the least water conservative and have the strongest overall tendency to increase ET with increasing VPD, while CSH are the most water conservative and have the strongest tendency to decrease ET with increasing VPD, as well as the strongest VPD dependence of response. Fig. 3 clearly shows,

according to our theory, that for all PFTs except for crops there is frequent occurrence of a negative (plant dominating) ET response to increases in VPD. Therefore, plants are able in most atmospheric conditions to reduce ET in response to increased VPD and thus to reduce water loss. To better illustrate this, the ranges of observed environmental VPDs at the FLUXNET sites are plotted parallel to the x-axis. For CSH and WSA, VPD is always less than $\mathrm{VPD}_{crit}$ (off scale) so that the plant response dominates in typical environmental conditions, emphasizing the water conservative strategy of those plants. For CRO on the

other hand, VPD is higher than $\mathrm{VPD}_{crit}$ for more than 50% of observations, emphasizing that those plants operate with an aggressive water usage strategy, are water intensive and were actually engineered for photosynthesis rather than water saving. For DBF, EBF, MF, GRA, and SAV more than half of the observed VPD are less than $\mathrm{VPD}_{crit}$, i.e. in conditions where plant response dominates. SAV has a more water conservative response than the forest, grass, and crop plan types, but still responds by increasing ET with increasing VPD for about a quarter of observations, due to the high aridity (VPD) of the SAV ecoclimate.

It is also important to note that for all PFTs, even when atmospheric demand dominates, ET response to VPD is still far more negative than it would be for potential evaporation $\partial PET/\partial VPD$, i.e. atmospheric demand only, emphasizing that there is still a strong regulation of evaporative flux by stomata and though the plant xylem. The sign term in the PET case would just be a constant ($\frac{c_p}{R_{air}} \approx 3.5$), which is far larger than any part of the curves for any PFT. Plants are always regulating water exchange from the land surface, even when they reach the limits of they ability to do so.

## 3.2   Functional Form of the Scaling Term

While the above discussion of the sign of $\frac{\partial ET}{\partial VPD}$ is important to answer our question of when ET response increases or decreases with VPD, understating the overall magnitude of the ET response is important to soil-plant-atmosphere water budgeting. So we now more closely examine the terms that affect how the sign term is scaled:

$$\frac{g_a \, P}{T(\Delta + \gamma)}. \tag{14}$$

$\frac{P}{T}$ is an air-density term, which varies little compared to aerodynamic conductance and Clausius-Clapeyron ($\Delta$). The psychometric constant ($\gamma$) is also relatively constant, so the scaling term should be primarily a function of aerodynamic conductance and temperature, through the Clausius-Clapeyron relationship $\Delta$. This is as expected, given that the aerodynamic conductance





represents the efficiency of exchange between the surface and the atmosphere. As aerodynamic conductance increases, any plant response will be communicated more strongly to the atmosphere (and vice-versa).

$\Delta$'s presence in the scaling term also matches physical intuition. $\Delta$ (and also the approximately constant $\gamma$) control the efficiencies with which surface energy is converted to latent and sensible heat (Monteith et al., 1965). The functional from of

$\Delta$ will be the same across PFTs, but the temperature range may vary slightly. In contrast, aerodynamic conductance will vary strongly with PFT due to the importance of surface roughness for aerodynamic conductance. So most of the differences in scaling between PFT should be in the aerodynamic conductance term.

The control of the scaling term variability between PFTs by aerodynamic conductance is confirmed by data (Fig. 4). Differences between PFT are almost entirely due to differences in aerodynamic conductance, rather than differences in observed

temperature ranges. The scaling term for the tree PFTs (DBF, EBF, ENF, MF) is generally about double the scaling terms for other PFTs which have lower surface roughness and generally smaller aerodynamic conductance (GRA, CSH, CRO). The savannah (WSA, SAV) PFT's scaling is somewhere between GRA, CSH, and CRO, and DBF, EBF, ENF, and MF, due to higher variability and surface roughness.

Within each PFT, the scaling term variability is controlled both by environmental temperature and aerodynamic conductance

variability (Fig. 4). While the observed variability of the aerodynamic conductance contributes more to the scaling term variability than temperature, the temperature contribution is non-negligible. Specifically, the scaling term is generally larger at low temperatures when latent heat is relatively inefficient at moving energy away from the surface. This effect amplifies the role of aerodynamic conductance variability at low temperatures.

To summarize, variability between PFTs is mostly controlled by systematic differences in aerodynamic conductance, due

to differences in surface roughness between each PFT, and possibly to a lesser extent wind conditions. In contrast, variability within PFT is also controlled by temperature, through Clausius-Clapeyron. But, aerodynamic conductance variability generally impacts the scaling term more than temperature, even within PFTs.

### 3.3 Bulk statistics of ET response to VPD

In this section we consider direct observations of ET response with eddy-covariance data, while including uncertainty with the

$\sigma$ term (Sect. 2.2). These observational results of ET response (Table 4) largely confirm our theoretical analysis, presented in the previous sections. For all PFTs, mean ET response to increasing VPD is negative. However, ET response evaluated at the average of all variables (e.g. $\sigma, T, c_a, VPD$) is positive for CRO, and negative for all other PFTs. This difference in mean ET response as compared to the ET response at mean environmental conditions is due to the non-linear nature of the response, in which negative responses are generally larger magnitude than positive responses (Fig. 3). Therefore, both the mean ET

response as well as the ET response at mean environmental conditions matches our expectations from the theory (Sect. 3.1), with the exception that CRO observations are shifted more towards a negative ET response than we expect.

Regarding the frequency of negative and positive ET response, all PFTs exhibit a decreasing ET response with increasing VPD (physiologically controlled, water conservative response) for the majority of observations. The more water conservative PFTs generally exhibit higher frequency of negative ET response, especially when one factors in the distribution of envi-



**Figure 4.** Primary sources of variability for the scaling term, as a function of PFT. The 5th-95th percentile range of temperature is plotted at the 5th, 25th, 50th, 75th, and 95th percentiles of aerodynamic conductance, as observed for each PFT.





**Table 4.** Statistics of $\frac{\partial ET}{\partial VPD}$ as a function of PFT.

| PFT | $\overline{\frac{\partial ET}{\partial VPD}}$ | $\frac{\partial ET}{\partial VPD}(\overline{env})$ | fraction $\frac{\partial ET}{\partial VPD} < 0.$ |
|-----|------|------|------|
| CRO | -0.041 | 0.014 | 0.513 |
| DBF | -0.110 | -0.017 | 0.618 |
| EBF | -0.108 | -0.013 | 0.634 |
| SAV | -0.038 | -0.031 | 0.650 |
| GRA | -0.072 | -0.022 | 0.690 |
| MF | -0.131 | -0.070 | 0.711 |
| WSA | -0.085 | -0.070 | 0.766 |
| ENF | -0.180 | -0.102 | 0.776 |
| CSH | -0.250 | -0.183 | 0.943 |

ronmental VPD (e.g. SAV and WSA grow in more arid climates, Fig. 3). In general the bulk statistics match our theoretical expectations well, with the caveat that inclusion of uncertainty shifts crops towards a slightly more negative response to VPD, and shifts many of the other PFTs, which still exhibit a high frequency of negative ET response to VPD, towards more frequent occurrence of positive ET response than the theory and Fig. 3 would suggest. The bulk statistics motivate a more thorough

examination of the structure of uncertainty and more sophisticated validation of our theory's performance against observations.

### 3.4  Validation of theory at eddy-covariance sites

We now compare more sophisticated distributions of the observed response to our simplified theory (Sect. 3.1). The observed distribution of the sign term, as compared to what the theory would predict, is provided in Fig. 5. Our goal was to capture the leading order behavior of the ET dependence on VPD. Given the assumptions we made, and the uncertainties of flux tower

observations themselves, we expect a relatively large amount of noise when reproducing the derivatives of ET. However, the data largely reproduces our theoretical analysis.

This is particularly true for DBF and MF; the theory matches the leading order behavior of the function when uncertainty is included, and the observations match the theory with the addition of noise. The VPD dependence, given by the slope parameter ($g_1$), follows the median values of each bin. Perhaps most importantly, the x-intercept, and thus $VPD_{crit}$, matches nearly exactly

between the theory and the observations. Therefore the sign of the ET response to increases in VPD should be well matched, subject to the unavoidable constraints of noise, much of which comes from the observations themselves. The uncertainty is non negligible; there are many observations in each bin for which the the sign of observation is opposite the response predicted by the theory, but to leading order our theory matches the observations well.

While CSH has a much different functional form of the sign term than DBF and MF, CSH observations also match our

theory to leading order, albeit with a bit more variability as a function of VPD. Again, the VPD dependence mostly determined from the slope parameter ($g_1$) closely matches the medians in the observation bins. The VPD-independent, strongly negative response is also captured. For CSH, there is rare occurrence of observed positive ET response with VPD ($\approx 6\%$), even with

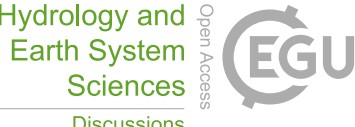

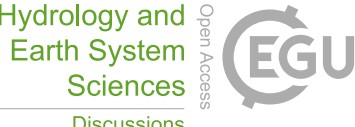

**Figure 5.** Comparison of the sign term with model uncertainty included (box plots) to the sign term as calculated with simplifying assumptions (blue line, as in Fig. 3). Each box plot corresponds to 5% of the data, and the 5-95% range of VPD is plotted.



uncertainty, so the sign of the observations almost always matches the sign of the theory, which states that ET response should always be negative.

Biases between the theory and observations are similar for ENF, WSA, CRO and SAV. At low VPD the theory and the observations match well. However, at high VPD (upper 10-20% of observations) the theory is biased slightly towards positive

response as compared to the observations (Fig. 5). For ENF this slight bias could be explained by a negative bias in $g_1$. However, for WSA, CRO, and SAV the observations in the highest VPD bins exhibit a downturn towards more negative response, which cannot be captured by the functional form of the sign term (Fig. 3). Therefore, to explain the observations one of the variables in the sign term must change at extreme VPD (upper 10-20%). The most likely candidate is $uWUE$, which we have assumed constant to meet our goal of developing intuition for leading order behavior, but might be expected to decrease at extremely

low SWC. This extremely low SWC should be correlated with high VPD in very dry environments through sensible heating increases (Gentine et al., 2016). The theory for EBF exhibits similar limitations as for ENF, WSA, and SAV, except a larger portion of observations are biased negative as compared to the theory ($\approx$ 35%, Fig. 5). However, in general, and specifically for non-extreme VPD (VPD $<\approx$70-90th percentiles), the theory matches the observations for the tree and savannah plant types well.

The theory for GRA suffers from similar, but much more severe, limitations as for CRO, WSA, and SAV. GRA observations are characterized by a consistent trend back towards negative ET response at higher VPD, which the functional form of our theory is incapable of accounting for. As compared to CRO, WSA and SAV, the divergence between the theory and the observations is far greater for GRA, biasing 40-50% of observations at higher VPD. In addition for the potential for soil moisture to alter $uWUE$, there are other sources of plant heterogeneity specific to GRA (and some extent CRO) that may alter $uWUE$

(or $g_1$) or invalidate other assumptions made in the methods section (Sect. 2.2). We do not account for variability in plant height and surface roughness, or differences in C3 vs C4 photosynthesis and water strategies, which we might expect to vary substantially across sites, years, and season for GRA. These deficiencies could largely explain the inability of our theory to exactly match the observations in croplands. For example, a superposition of sites with C3 plants (at low environmental VPD) and C4 plants (at high environmental VPD) would explain the observed shift back towards negative ET response at high VPD

when all sites are binned together, as in Fig. 5. We hypothesize that the theory would validate against observations much better if these sources of variability were accounted for, at a cost of increased complexity and analytical opacity.

While the above discussion shows that our theory has some limitations when applied to some PFTs, especially for grasslands, it does well for DBF, MF and CSH PFTs, and captures the response at non-extreme VPD (VPD $<\approx$ 70th-90th percentiles) for ENF, EBF, CRO, WSA, and SAV. In general, the leading order behavior observed in the data is captured by the theory.

Departures between our theory and observations, specifically at high VPD, could be explained and conceptualized with shifts in $g_1$ and/or $uWUE$ due to site-specific plant type variability (e.g. more arid-adapted ecosystems at more arid sites), or temporal variability for some environmental conditions (e.g. decreases in $uWUE$ at extremely low soil moisture). To focus on general behavior and develop intuition for PFT-scale response, we have ignored these sources of variability in the present analysis. However, any site-specific or temporal plant functional variability that can be conceptualized with shifts to $g_1$ and $uWUE$ can




be analyzed with our framework. This opens a door to future analyses in which plant behavior in anomalous conditions can be explained and analyzed using Eq. (7).

### 3.5 Observed ET response to VPD

Most of the results presented so far focus on the sign term, so now we turn to observations of ET response with the scaling
term included (Fig. 6). Until now, in the interest of developing leading order intuition for ET response, and to be conservative in our acknowledgment of model and observational error, we've considered $\sigma$ variability to be a measure of uncertainty. An alternative viewpoint is that $\sigma \cdot uWUE$ represents spatial and temporal variability of uWUE, which may be expected within bounds (see Table 2, also Zhou et al., 2015). This is a less conservative view; some of the $\sigma \cdot uWUE$ variability will be due to model and observational error, so by viewing $\sigma \cdot uWUE$ as *real* variability we run the risk of mistaking noise for signal.
However, the advantage of this viewpoint is that, from Zhou et al. (2014), we have very high confidence that uWUE fit at the hourly timescale (as we do with $\sigma \cdot uWUE$) correctly captures the relationship between ET, GPP, and VPD, and that our form of PM introduces minimal error with its use of uWUE.

Therefore, with the caveat that some of the signal presented in Fig. 6 may in fact be noise, we can interpret the observed distribution of ET response to VPD. In general, the observed response matches the intuitive theory. ET response to VPD shifts
towards positive values as VPD increases (atmospheric demand dominating). CRO exhibit the highest occurrence of positive ET response, and the observations confirm that CRO are the most water intensive. CSH are the most water conservative, with a strong negative response. DBF, EBF, ENF, MF, SAV and WSA are also water conservative, but show some occurrence of positive ET response to VPD, particularly at higher observed VPD, as the theory predicts. GRA, while generally water conservative, does not match the theory well, with increasing frequency of *negative* response at high VPD. This is as expected,
given previous discussion on how we might expect more inter-site and inter-year variability for GRA (Sect. 3.4).

Fig. 6 also includes the impact of the scaling term. For a given VPD, the magnitude of ET response does not vary strongly with temperature, confirming that any impact of the scaling term on the magnitude of the response is primarily due to changes in aerodynamic conductance. Intuitively this is reasonable; aerodynamic conductance will control how dominant balances at the land surface are communicated to the boundary layer. While $\Delta$ controls the efficiency of energy conversion to latent heat,
it appears this is a second order term, relative to $g_a$, for scaling ET response.

As with Fig. 5, Fig. 6 matches our expectation based on simplified theory. The sign term is most strongly scaled by $g_a$, and in general the occurrence of positive ET response increases as VPD increases. The willingness with which a given plant type evolved to use water dictates the occurrence of positive versus negative ET response. Water conservative ecosystems are highly effective at mitigating the effects of atmospheric demand, and can store water for later use by reducing ET in response
to increasing VPD.

### 3.6 Limitations of theory: very dry soil moisture conditions

In formulating our theory with Penman Monteith, we implicitly did not account for very dry soil moisture conditions. For the majority of environmental conditions observed at the eddy covariance sites used here, soil conditions were not extremely dry so





**Figure 6.** Scatter plots of observed $\frac{\partial ET}{\partial VPD}$, including $\sigma$ variability, as a function of PFT, temperature and VPD. Please note differences in the colorbar scale.





that we could assume a constant uWUE and $g_1$. We posit that ecosystems will generally optimize to host plants living in conditions which they evolved for. However, in extreme conditions and drought scenarios soil water content (SWC) could become the limiting factor for ET response to VPD, which our theory does not account for. In addition, low soil moisture conditions themselves increase VPD through land-atmosphere feedback (Bouchet, 1963; Morton, 1965; Brutsaert, 1999; Ozdogan et al.,
2006; Salvucci and Gentine, 2013; Gentine et al., 2016; Berg et al., 2016).

Within our framework, any systematic bias due to the failure to account for SWC's effects in very dry conditions should manifest itself in a functional relationship between $\sigma$, our uncertainty measure, and SWC, which is observed at the FLUXNET sites. Examining the relationship between $\sigma$ and SWC will test to what extent our theory breaks down in very dry soil moisture conditions.

If we again view $\sigma \cdot uWUE$ as, in addition to a measure of uncertainty, a short time-scale observation of uWUE, we would expect $\sigma \cdot uWUE$ to decrease at low SWC. If $\sigma \cdot uWUE$ is a very strong function of SWC, then our theory should be conditioned more strongly on well watered soil conditions. If $\sigma \cdot uWUE$ is weakly a function of SWC, then our theory is more universal and independent of soil moisture conditions.

Indeed, for all PFTs, there is some slight dependence of $\sigma \cdot uWUE$ on SWC, especially at low SWC (Fig. 7). The portion
of observations for which our theory is biased by SWC-limitations varies by PFT, due to the nonlinear threshold at which soil moisture availability limits plant function. For CRO, MF and DBF, soil water content only matters for about the lowest 5% of observations (each box is 5% of observations in Fig. 7). And, CRO, relative to DBF, has a very weak dependence on soil moisture, which is reflective of the high likelihood that CRO sites are optimally irrigated and not water stressed, suggesting that observed departures from the theory for CRO are due to other factors (e.g. different photosynthesis pathways; C3/C4)
than soil moisture. For ENF, EBF, SAV, and WSA, systematic SWC-induced biases in $\sigma \cdot uWUE$ emerge in about the lowest 20-30% of SWC conditions, although large variability in the SWC-($\sigma \cdot uWUE$) relationship hampers interpretation for WSA. CSH presents a special case: the limited number of sites (2) preclude comment on the exact PFT-wide relationship.

In contrast to ENF, CRO, and DBF, for GRA the relationship between $\sigma \cdot uWUE$ and SWC is more linear and affects a greater portion of observations; about 60% of observations. Clearly, for GRA soil water frequently impacts plant function and
alters ET response, and our theory is limited for the majority of environmental conditions. It is therefore not surprising that our theory tested poorly against the data for GRA, relative to to the other PFTs (Sect. 3.4). For all other PFTs occurrence of soil moisture impacts was rare enough to not manifest itself in bulk statistics and figures.

The observed dependence of $\sigma \cdot uWUE$ on SWC for GRA would explain the deficiencies of our theory compared to the observations in Fig. 5, specifically the trend back towards negative ET response at high VPD. Aforementioned feedbacks
between the land surface and the atmosphere, which are not accounted for due to our focus on the one-way response of the land surface to atmospheric conditions, would cause high VPD to be correlated with low SWC (Gentine et al., 2016; Berg et al., 2016). So, at high VPD observations of low SWC are more likely, and this low SWC causes a lower uWUE (Fig. 7). The lower uWUE at low SWC/high VPD then leads to the observed downturn towards decreasing ET with increasing VPD, and the deviation between our theory (based on a constant uWUE assumption) and observations. It is also perhaps not coincidental that
the portion of $\sigma \cdot uWUE$ affected by low soil moisture observations is similar to the portion of observations that do not match







**Figure 7.** $\sigma \cdot uWUE$ as a function of PFT and SWC. Each box plot represents 5% of all observations. Note that the highest 10% of SWC observations are excluded to better resolve variability in the much more narrow bins at lower SWC (e.g. SWC has long tails at high values).




our theory at high VPD. Indeed, this may suggest that for all PFTs except for GRA, coupling and feedbacks between SWC, VPD and plant function are relatively rare. Future research will explore these relatively rare feedbacks in extreme conditions, which due to analytical intractability will require more opaque numerical analysis of many more complex processes, including boundary layer growth and state and their relationship to surface layer coupling and free-tropospheric lapse rates and humidity.

To summarize, our theory is limited by its inability to account for soil water impacts on land surface response, and feedbacks between SWC and VPD. Fortunately, for most PFTs SWC's effect on ET is relatively rare ($<$30% of observations) and does not manifest itself in the majority of observations and bulk statistics. However for GRA, SWC decreases water use efficiency for the majority of the observations. Soil moisture effects explain the deficiencies of our theory in Sect. 3.4, particularly for GRA. By conceptualizing SWC effects as a change in $uWUE$ (and/or $g_1$), it will be possible for future analysis to explore the

importance of soil moisture on plant response to VPD, and feedbacks between plant function, SWC, and VPD.

### 3.7   Functional form of ET dependence on VPD and its relation to the VPD exponent

The theory described in Sect. 3.1 indicates that for a given $uWUE$ and $g_1$, the ET dependence on VPD should be concave upward, which is confirmed by eddy covariance data across most PFTs. In other words, there should be some local minimum in ET at a critical VPD$_{crit}$, assuming the scaling and plant terms (e.g. aerodynamic conductance, $\Delta$, $g_1$ and $uWUE$) are held

fixed. This result warrants further investigation, because to our knowledge no one has derived the theoretical ecosystem-scale relationship between ET and VPD while controlling for other environmental conditions. In particular, from personal communication, there is an apparent lack of consensus over whether the shape of the ET-VPD curve should be concave upward (our result) or concave downward in the absence of dramatic water stress. Given that understanding the ET-VPD relationship of the one-way plant response is fundamental to hypothesizing about any feedbacks between the land surface and the atmosphere, we

analyze why our derived ET-VPD relationship is concave upward, particularly with respect to the exponent of VPD dependence in $uWUE$ and the Medlyn unified stomatal conductance model.

There is a theoretical basis for the square root VPD dependence in both the stomatal conductance model and $uWUE$ based on the assumption that stomata behave to maximize carbon gain while minimizing water loss, which observations also generally support (Lloyd, 1991; Medlyn et al., 2011; Lin et al., 2015; Zhou et al., 2014, 2015; Medlyn et al., 2017). However, some purely

empirical results that fit the exponent of the VPD dependence to data have shown that it may vary slightly from 1/2, suggesting that stomata, as well as ecosystem-scale quantities based on stomata theory, may not always function optimally (Zhou et al., 2015; Lin et al., 2018). Specifically with regards to $uWUE$, one would not expect that this ecosystem scale WUE quantity will respond to VPD exactly analogously to stomata. Direct soil evaporation's contributions to ET should shift the exponent of the VPD dependence, especially at conditions of low GPP when we would expect a systematically larger portion of direct

soil evaporation contributions to ET, because we would also expect lower amounts of transpiration at low GPP. Zhou et al. (2015)'s results corroborate this: they found a mean empirically fit exponential VPD dependence of 0.55, varying slightly from the theoretically optimal value of of 0.5 for AmeriFlux sites. Interpreting Lin et al. (2018)'s results, which also show variance in the empirical exponent of the VPD dependence of the stomatal conductance model, is more difficult as Lin et al. (2018) do not handle GPP/A dependence of stomatal conductance in a directly analogous manner to the optimal theory in Medlyn et al.





(2011) and Medlyn et al. (2017). Regardless, given that these recent results on the relationship between VPD, GPP, and ET (Medlyn et al., 2011; Zhou et al., 2014, 2015; Medlyn et al., 2017) form the backbone of our analysis and are what allowed us to derive an explicit ET expression for the first time (Eq. (7)), we will analyze if and how assumptions about the exponent of the VPD dependence impacts the shape of the ET-VPD dependence. This analysis is also important to understand whether the

choice of stomatal conductance model alters the fundamental behavior of the ET-VPD relationships, as many commonly used models utilize a VPD exponent other than the 1/2 suggested by optimal theory (e.g. Leuning, 1990, which uses an exponent of 1).

By introducing $n$ and $m$ we can free our stomatal conductance model from assumptions about VPD dependence:

$$g_s = \frac{RT}{P} 1.6 \left(1 + \frac{g*}{VPD^m}\right) \frac{*WUE\ ET}{c_a\ VPD^n}, \tag{15}$$

where:

$$*WUE = \frac{GPP}{ET} VPD^n,$$

and $g*$ is a generic slope parameter of units $VPD^m$. To determine how the exponent $n$ and $m$ alter the shape of the ET-VPD dependence we find the roots of the second derivative of ET, using Eq. (15) for stomatal conductance ($g_s$), with respect to VPD:

$$\frac{\partial^2\ ET}{\partial\ VPD^2} = 0 \quad \forall \quad \frac{VPD^m}{g*} = \frac{m\left(m - 2n - \sqrt{m^2 - 4mn + 2m - 4n^2 + 4n + 1} + 1\right)}{2n(n-1)} - 1. \tag{16}$$

With this result we have defined the family of curves separating concave up from concave down ET solutions (Fig. 8). These curves are only functions of the exponent of the VPD dependence and a quantity we call non-dimensional VPD ($VPD^m/g*$). Several important relations reveal themselves from Eq. (16):

- For optimal behavior (n, m = 1/2) the ET-VPD curve will be concave up regardless of the magnitude of the plant constants
$g_1$ and $uWUE$. Therefore, the general concave up nature of our results, given an assumption of optimal behavior, is insensitive to plant type.

- For all physically possible exponents of VPD dependence $(n, m)$, whether the solution is concave up or concave down does not depend on $uWUE$.

- In general, increasing the exponent of VPD dependence increases the likelihood of a concave down result. Addition-
ally, as the exponent of VPD dependence increases from the optimum value of 1/2, whether the curve is concave upward or concave downward becomes a function of the plant specific slope parameter $g_*$, through non-dimensional VPD ($VPD^m/g_*$). Because the exponent of the VPD dependencies is capable of altering the fundamental shape of ET-VPD dependence, future research investment in understanding the exact VPD dependence of stomatal conductance, and further reconciliation of empirical and theoretical stomatal and ecosystem behavior should be prioritized.

While it is possible that in the future some other form of VPD dependence is derived, at present Medlyn et al. (2011) and Zhou et al. (2014) firmly established n=m=1/2 as the most likely candidate given current theory and empirical data. Additionally,





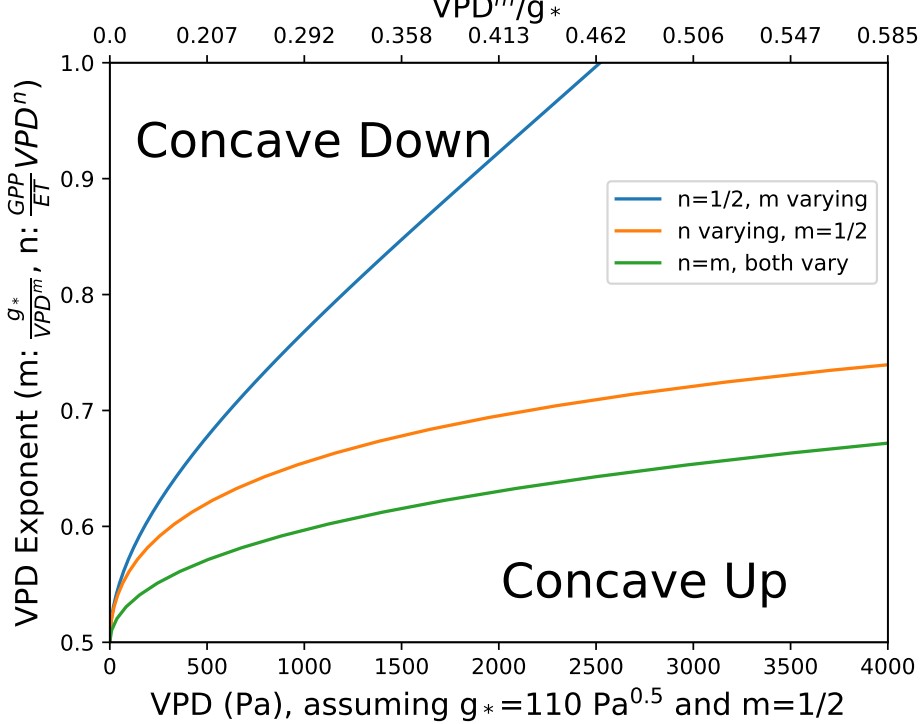

**Figure 8.** Solutions corresponding to inflection points between concave up and concave down ET-VPD curves (Eq. (16)) for three specific scenarios. Solutions are defined in terms of a non-dimensional VPD ($VPD^m/g_*$), but to aide physical interpretation the horizontal axis is additionally provided in terms of dimensionalized VPD assuming $m = 1/2$ and $g_* = 110\ Pa^{1/2}$ (average of all PFT $g_1$). The vertical axis has a different interpretation depending on the solution curve. For the blue line ($m$ varying), it corresponds to $m$, for the orange line ($n$ varying) it corresponds to $n$, and for the green line it corresponds to the value of both $n$ and $m$ ($n = m$). Regions of the parameter space that correspond to concave up and concave down results are labeled.

we argue that a concave up result matches physical intuition more than a concave down result. Plants must maintain nutrient and sugar transport through the phloem and xylem. To accomplish this, stomata must remain slightly open (De Schepper et al., 2013; Nikinmaa et al., 2013; Ryan and Asao, 2014). Furthermore, even if complete stomatal closure were possible, cuticular water loss and [at the ecosystem-scale] direct soil evaporation are still sources of ET which increase with VPD, independent

5    of stomatal closure. Therefore, in the limit as VPD becomes large and we assume plants are exercising all strategies to reduce ET, any further increase in VPD should result in an increase in ET through cuticular water loss and/or direct soil evaporation. This inevitable transition from conditions when stomata respond strongly to VPD to conditions when stomata response is asymptoting towards full closure would cause a concave up ET-VPD curve, which is matched by the theory. In short, plant response becomes more limited as VPD increases, while atmospheric demand monotonically increases with VPD, leading to

10   the result that atmospheric demand dominates plant response when atmospheric demand is high.





This analysis allows us to understand the theoretical shape of the ET response to VPD with environmental conditions held fixed. Accomplishing this with purely statistical methods applied to flux observations would be very difficult, given the relatively fast time scale of plant response and the non-stationarity of [solar forced] environmental conditions over the relatively coarse (half hourly) flux estimates (which is required to obtain robust eddy-covariance statistics). Our results on the shape of

the ET-VPD curve with environmental conditions held fixed can be built upon with future work examining how changes in VPD and environmental conditions (e.g. soil water storage) feedback upon one another. In the soil water storage example, over very long time scales extremely high VPD perturbations coupled with no precipitation could result in decreases in soil water storage such that water becomes limiting. This could be represented by an extension of our framework in which uWUE is allowed to decrease with decreasing SWC, as observed in Sect. 3.6. Here, we focus our results by assuming constant PFT-

wide conditions to build baseline intuition for ET-VPD dependence. For most PFTs, the theory with plant function held fixed matches the leading order behavior of the observations where plant function varies (Sect. 3.4).

## 4 Conclusions

We derived a new form of Penman Monteith using the concept of semi-empirical optimal stomatal regulation (Lin et al., 2015; Medlyn et al., 2011) and near constant uWUE (Zhou et al., 2015) to remove the implicit dependence of stomatal conductance

on GPP and ET. With our new form of Penman Monteith we developed a theory for when an ecosystem will tend to reduce or increase ET with increasing VPD, which we evaluated against a range of eddy-covariance data spanning different climates and plant functional types. The goal was to capture the leading order behavior of the system to gain some intrinsic knowledge for its behavior. This intuition can be used to disentangle land atmosphere feedbacks in more complicated scenarios, and will aid interpretation of observations and sophisticated models.

Our theory suggests that for a majority of environmental conditions, plants will tend to conserve water and reduce ET with increasing VPD. Stomatal regulation and plant physiological response strongly regulate ET, and this regulation varies by PFT. CROs are the least water conservative, while DBF, EBF, SAV, GRA, MF, WSA, ENF and CSH are progressively more water conservative (more likely to reduce ET in response to increasing VPD). SAV and WSA exhibit positive ET response to VPD not necessarily because of poor water conservation strategies relative to other PFTs, but because of greater occurrence of high

atmospheric demand (VPD) relative to other PFTs. Observations of ET response to VPD exhibit the same general behavior as the theory, with ET response becoming more positive (atmospheric demand dominating) as environmental VPD increases within a PFT, and more negative for PFTs that are adapted to arid conditions and prioritize water conservation over primary production.

Our paper builds important intuition for how plants respond to VPD perturbations. We show that given optimal stomatal

function and fixed environmental conditions, the ET-VPD dependence is theoretically concave upward, with ET increasing with increasing VPD as VPD increases past some critical value (Table 3). However future research should focus on fully understanding the form of stomatal VPD dependence, as this result is sensitive to the exponent of VPD dependence, which we currently believe is 1/2 (Medlyn et al., 2011; Zhou et al., 2014). Indeed, this sensitivity to the exponent of VPD dependence





is an important result itself: land surface models, including those used in earth system models for climate forecasts, employ different assumptions about the exponent of VPD dependence in stomatal conductance (e.g., Ball et al., 1987; Leuning, 1990; Medlyn et al., 2011), and these assumptions can fundamentally change the relationship between ET and VPD from one that is concave upward (local minimum in ET) to one that is concave downward (local maximum in ET).

Our results are also applicable to understanding the impact of expected increases in VPD induced by global change. Plant physiological responses to direct $CO_2$ effects (e.g., Swann et al., 2016; Lemordant et al., 2018) receives more attention than physiological response to indirect effects like increased VPD. Here, we provide broad PFT-focused results showing a likely decrease in ET in response to positive VPD perturbations (atmospheric drying), which is consistent with recent observational analysis (e.g., Rigden and Salvucci, 2017). Feedbacks between the land and the atmosphere may alter the net response to a

long-timescale global VPD perturbation, but our focus on the one way plant response to a VPD perturbation in the atmospheric boundary layer is an important first step to disentangling such feedbacks, both in observations and model simulations of the present and future. By removing Penman Monteith's dependence on implicit relationships between GPP, VPD, and ET, we allow for explicit, accurate future analysis of plant-VPD feedbacks in the atmospheric boundary layer (Eq. (7)). Our approach can be extended to examine varying plant response to more nuanced consideration of plant type and climate. Any plant phys-

iological heterogeneity or feedback that can be conceptualized with shifts in $g_1$ (e.g. Lin et al., 2015; Medlyn et al., 2017) and/or uWUE (e.g. Zhou et al., 2014) are representable within our framework, which opens the door for a hierarchy of more sophisticated climate- and plant-specific analysis of ET sensitivity to environmental variables (including VPD). We argue that such simplified conceptual frameworks are critical tools for disentangling land-atmosphere feedbacks at various scales, from diurnal to seasonal and beyond, and to characterize ET response in a warmer, atmospherically drier, and enriched $CO_2$ world.

*Code and data availability.* All code and data used in this analysis, including those used to generate the figures and tables, are publicly available at https://github.com/massma/climate_et

## Appendix A:  FLUXNET2015 sites

Table A1: Metadata and citations for flux sites used in this analysis. All data are gathered from www.fluxdata.org, and citations are aggregated using tools available at https://github.com/trevorkeenan/FLUXNET_citations.

| Site | PFT | Lat | Lon | Clim[1] | Period | References |
|---|---|---|---|---|---|---|
| AT-Neu | GRA | 47.1167 | 11.3175 | Unk | 2002-2012 | Wohlfahrt et al. (2008) |
| AU-ASM | ENF | -22.2830 | 133.2490 | Unk | 2010-2013 | Cleverly et al. (2013) |
| AU-Cpr | SAV | -34.0021 | 140.5891 | Unk | 2010-2014 | Meyer et al. (2015) |
| AU-DaP | GRA | -14.0633 | 131.3181 | Aw | 2007-2013 | Beringer et al. (2011) |
| AU-DaS | SAV | -14.1593 | 131.3881 | Aw | 2008-2014 | Hutley et al. (2011) |
| AU-Dry | SAV | -15.2588 | 132.3706 | Unk | 2008-2014 | Cernusak et al. (2011) |



| | | | | | | |
|---|---|---|---|---|---|---|
| AU-Gin | WSA | -31.3764 | 115.7138 | Unk | 2011-2014 | Beringer et al. (2016) |
| AU-How | WSA | -12.4943 | 131.1523 | Aw | 2001-2014 | Beringer et al. (2007) |
| AU-Rig | GRA | -36.6499 | 145.5759 | Unk | 2011-2014 | Beringer et al. (2016) |
| AU-Stp | GRA | -17.1507 | 133.3502 | Unk | 2008-2014 | Beringer et al. (2011) |
| AU-Tum | EBF | -35.6566 | 148.1517 | Cfb | 2001-2014 | Leuning et al. (2005) |
| AU-Whr | EBF | -36.6732 | 145.0294 | Unk | 2011-2014 | McHugh et al. (2017) |
| AU-Wom | EBF | -37.4222 | 144.0944 | Unk | 2010-2012 | Hinko-Najera et al. (2017) |
| BE-Lon | CRO | 50.5516 | 4.7461 | Cfb | 2004-2014 | Moureaux et al. (2006) |
| BE-Vie | MF | 50.3051 | 5.9981 | Cfb | 1996-2014 | Aubinet et al. (2001) |
| BR-Sa3 | EBF | -3.0180 | -54.9714 | Am | 2000-2004 | Wick et al. (2005) |
| CA-Qfo | ENF | 49.6925 | -74.3421 | Dfc | 2003-2010 | Bergeron et al. (2007) |
| CA-SF1 | ENF | 54.4850 | -105.8176 | Dfc | 2003-2006 | Mkhabela et al. (2009) |
| CA-SF2 | ENF | 54.2539 | -105.8775 | Dfc | 2001-2005 | Mkhabela et al. (2009) |
| CH-Cha | GRA | 47.2102 | 8.4104 | Unk | 2005-2014 | Merbold et al. (2014) |
| CH-Dav | ENF | 46.8153 | 9.8559 | Unk | 1997-2014 | Zielis et al. (2014) |
| CH-Fru | GRA | 47.1158 | 8.5378 | Unk | 2005-2014 | Imer et al. (2013) |
| DE-Geb | CRO | 51.1001 | 10.9143 | Unk | 2001-2014 | Anthoni et al. (2004) |
| DE-Gri | GRA | 50.9500 | 13.5126 | Cfb | 2004-2014 | Prescher et al. (2010) |
| DE-Hai | DBF | 51.0792 | 10.4530 | Unk | 2000-2012 | Knohl et al. (2003) |
| DE-Kli | CRO | 50.8931 | 13.5224 | Cfb | 2004-2014 | Prescher et al. (2010) |
| DE-Lkb | ENF | 49.0996 | 13.3047 | Unk | 2009-2013 | Lindauer et al. (2014) |
| DE-Obe | ENF | 50.7867 | 13.7213 | Cfb | 2008-2014 | – |
| DE-Seh | CRO | 50.8706 | 6.4497 | Unk | 2007-2010 | Schmidt et al. (2012) |
| DE-Tha | ENF | 50.9624 | 13.5652 | Cfb | 1996-2014 | Grünwald and Bernhofer (2007) |
| DK-Sor | DBF | 55.4859 | 11.6446 | Unk | 1996-2014 | Pilegaard et al. (2011) |
| FI-Hyy | ENF | 61.8474 | 24.2948 | Unk | 1996-2014 | Suni et al. (2003) |
| FI-Sod | ENF | 67.3619 | 26.6378 | Unk | 2001-2014 | Thum et al. (2007) |
| FR-Gri | CRO | 48.8442 | 1.9519 | Cfb | 2004-2013 | Loubet et al. (2011) |
| FR-LBr | ENF | 44.7171 | -0.7693 | Unk | 1996-2008 | Berbigier et al. (2001) |
| IT-Col | DBF | 41.8494 | 13.5881 | Unk | 1996-2014 | Valentini et al. (1996) |
| IT-Cpz | EBF | 41.7052 | 12.3761 | Unk | 1997-2009 | Garbulsky et al. (2008) |
| IT-Lav | ENF | 45.9562 | 11.2813 | Unk | 2003-2014 | Marcolla et al. (2003) |
| IT-MBo | GRA | 46.0147 | 11.0458 | Unk | 2003-2013 | Marcolla et al. (2011) |
| IT-Noe | CSH | 40.6061 | 8.1515 | Unk | 2004-2014 | Papale et al. (2014) |
| IT-Ren | ENF | 46.5869 | 11.4337 | Unk | 1998-2013 | Montagnani et al. (2009) |



| IT-Ro2 | DBF | 42.3903 | 11.9209 | Unk | 2002-2012 | Tedeschi et al. (2006) |
| IT-SRo | ENF | 43.7279 | 10.2844 | Unk | 1999-2012 | Chiesi et al. (2005) |
| IT-Tor | GRA | 45.8444 | 7.5781 | Unk | 2008-2014 | Galvagno et al. (2013) |
| NL-Loo | ENF | 52.1666 | 5.7436 | Unk | 1996-2013 | Moors (2012) |
| RU-Fyo | ENF | 56.4615 | 32.9221 | Unk | 1998-2014 | Kurbatova et al. (2008) |
| US-AR1 | GRA | 36.4267 | -99.4200 | Dsa | 2009-2012 | Raz-Yaseef et al. (2015) |
| US-AR2 | GRA | 36.6358 | -99.5975 | Dsa | 2009-2012 | Raz-Yaseef et al. (2015) |
| US-ARM | CRO | 36.6058 | -97.4888 | Cfa | 2003-2012 | Fischer et al. (2007) |
| US-Blo | ENF | 38.8953 | -120.6328 | Csa | 1997-2007 | Goldstein et al. (2000) |
| US-KS2 | CSH | 28.6086 | -80.6715 | Cwa | 2003-2006 | Powell et al. (2006) |
| US-MMS | DBF | 39.3232 | -86.4131 | Cfa | 1999-2014 | Dragoni et al. (2011) |
| US-Me2 | ENF | 44.4523 | -121.5574 | Csb | 2002-2014 | Irvine et al. (2008) |
| US-NR1 | ENF | 40.0329 | -105.5464 | Dfc | 1998-2014 | Monson et al. (2002) |
| US-Ne1 | CRO | 41.1651 | -96.4766 | Dfa | 2001-2013 | Verma et al. (2005) |
| US-Ne2 | CRO | 41.1649 | -96.4701 | Dfa | 2001-2013 | Verma et al. (2005) |
| US-Ne3 | CRO | 41.1797 | -96.4397 | Dfa | 2001-2013 | Verma et al. (2005) |
| US-SRG | GRA | 31.7894 | -110.8277 | Bsk | 2008-2014 | Scott et al. (2015) |
| US-SRM | WSA | 31.8214 | -110.8661 | Bsk | 2004-2014 | Scott et al. (2009) |
| US-Syv | MF | 46.2420 | -89.3477 | Dfb | 2001-2014 | Desai et al. (2005) |
| US-Ton | WSA | 38.4316 | -120.9660 | Csa | 2001-2014 | Baldocchi et al. (2010) |
| US-Var | GRA | 38.4133 | -120.9507 | Csa | 2000-2014 | Ma et al. (2007) |
| US-WCr | DBF | 45.8059 | -90.0799 | Dfb | 1999-2014 | Cook et al. (2004) |
| US-Wkg | GRA | 31.7365 | -109.9419 | Bsk | 2004-2014 | Scott et al. (2010) |
| ZA-Kru | SAV | -25.0197 | 31.4969 | Unk | 2000-2010 | Archibald et al. (2009) |
| ZM-Mon | DBF | -15.4378 | 23.2528 | Unk | 2000-2009 | Merbold et al. (2009) |

[1] Köppen Climate classification.

*Author contributions.* AM: formal analysis; methodology; software; validation; visualization; writing - original draft. PG: conceptualization; methodology; supervision; writing - review & editing. CL: data curation; writing - review & editing.

*Competing interests.* We declare no competing interests.





*Acknowledgements.* This work used eddy covariance data acquired and shared by the FLUXNET community, including these networks: AmeriFlux, AfriFlux, AsiaFlux, CarboAfrica, CarboEuropeIP, CarboItaly, CarboMont, ChinaFlux, Fluxnet-Canada, GreenGrass, ICOS, KoFlux, LBA, NECC, OzFlux-TERN, TCOS-Siberia, and USCCC. The ERA-Interim reanalysis data are provided by ECMWF and processed by LSCE. The FLUXNET eddy covariance data processing and harmonization was carried out by the European Fluxes Database Cluster, AmeriFlux Management Project, and Fluxdata project of FLUXNET, with the support of CDIAC and ICOS Ecosystem Thematic Center, and the OzFlux, ChinaFlux and AsiaFlux offices. We would like to thank Dr. Trevor Keenan for providing tools for citation of the FLUXNET2015 dataset, available at https://github.com/trevorkeenan/FLUXNET_citations. This material is based upon work supported by the National Science Foundation Graduate Research Fellowship under Grant No. DGE 16-44869.





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
