# Peer review of "When does vapor pressure deficit drive or reduce evapotranspiration?"

_Hydrology and Earth System Sciences, 2018_

## Referee Comment (RC1) · Anonymous Referee #1 · 8 Dec 2018

**Review of "When does vapor pressure deficit drive or reduce evapotranspiration?" submitted to HESS, by Adam Massmann, Pierre Gentine, and Changjie Lin**

The authors address an extremely compelling problem in this manuscript – the question of whether increasing vapor pressure deficit will lead to decrease in evapotranspiration (due to stomatal regulation to reduce water loss) or increase in evapotranspiration (due to increased atmospheric demand). Answering this question will advance our capabilities to understand and predict ecosystem-level differences in water uptake in response to warming, and thus deserve attention and effort.

Methodologically, the authors employ a mix of models from (1) land-atmosphere coupling (Penman-Monteith, based on radiation balance and surface and air conductance of water vapor into atmosphere), (2) leaf-scale optimal stomatal conductance theory (from Medlyn et al. 2011) that relates stomatal conductance Gs to GPP, VPD, and a water use efficiency term $g_1$, and (3) an empirically-derive relationship between ecosystem scale GPP, ET, VPD, and an "underlying water use efficiency" uWUE:

(1) Penman-Monteith:  ET = $f_1$(VPD, Gs)
(2) Optimal stomatal conductance (Medlyn et al. 2011):  Gs = $f_2$($g_1$, VPD, GPP)
(3) Underlying water use efficiency (Zhou et al. 2014):  uWUE = $f_3$(GPP, ET, VPD)

**Together, they present a set of three closed equations that can be used to eliminate the dependence on GPP, and relate ET as a function of VPD, $g_1$, and uWUE only.**

All the results follow from the assumptions set out in the derivation of the model, so I will focus my comments mostly on the derivations. I am mainly concerned about two issues:

1. Compounding uncertainties in parameters across each one of the equations used
2. Interpretation and attribution of observed effects to plant physiological responses

**1. Compounding uncertainties**

A key premise of this work is that g1 and uWUE should exhibit greater variation across ecosystem types than within, thus distinguishing ecosystem types and responses from each other. In trying to see whether this premise is valid, I went back to check on the works of Medlyn et al. (2017) and Zhou et al. (2014), who have already derived values of g1 and uWUE respectively in their own works, and examined the range of previously derived parameter values and compared it with the values fitted using the current model.

The results were unconvincing. The ranking of these parameter values across ecosystems are not preserved. This is shown in Table 2 of the manuscript … whereas Zhou et al. (2014) predicted the crop types (CRO) and evergreen needleleaf forest (ENF) to have the highest uWUE values of all ecosystem types analyzed, the model in the current manuscript predicted both crops and needleleaf forests to only have a moderate, middle-of-the-road uWUEs. The

current model instead predicts the highest g1 values for crop types, but those had the lowest mean g1 values as predicted by Medlyn et al. (2017) (Figure 7; C3C and C4C). Thus, the rankings of g1 and uWUE values were found to be inconsistent across all these studies.

How should we interpret this? I'm not sure the authors offer any insights into this question. Certainly, they have acknowledged the existence of uncertainty around the temporal and spatial variations of uWUE and g1 by introducing an uncertainty parameter *sigma*. But given the potential sources of error in working with high resolution ecosystem scale data, and the range of values that both parameters can take on *within* the same ecosystem type, I wonder if it might be more effective to think about the relationship between ET and VPD more probabilistically as a function of pdfs – rather than point estimates – of g1 and uWUE.

The questions still remain, though, that according to Medlyn et al. (2017) Figure 7, the degree of variability in g1 *within* ecosystem types and using different methods of derivation (via leaf, isotope, or flux data), can be as high as variability in g1 *across* ecosystem types. The authors do not seem to have addressed this issue of *within-PFT* range in g1 and uWUE, and without it, I found it very difficult to interpret what these parameter values mean and what they could be useful for. If indeed these discrepancies arise from the selection of sites and/or time periods, then how sensitive would the results be to these choices of analyses?

**2. Attribution to physiological responses**

I am also concerned about the role of g1 and uWUE in masking potential contribution of soil moisture to ET. Essentially, I don't think it's correct to say that g1 and uWUE are attributes of the PFT *only*, which is another key premise of the authors' interpretation of the results.

There now exist a substantial body of work that suggest that *lambda* (the marginal water cost of leaf carbon used as the Lagrange multiplier in the calculus of variations for optimizing stomatal conductance) varies under water deficient / droughted conditions (see Makala et al. 1996, Annals of Botany; Kirschbaum 1999, Ecological Modeling; and other references within Medlyn et al. 2011). **This means that g1 itself, which is a function of *lambda*, should vary under water-limited conditions.** This functional dependence of g1 on soil moisture is also supported by empirical works from AmeriFlux sites such as those from Novick et al. (2016). Medlyn et al. (2011) itself states that: "*It can be questioned whether the optimization criterion assumed here can still be said to be optimal if drought stress starts to threaten plant survival. It may be that the relationship given by Eqn (11)* (the one used in the current manuscript) *will break down as soil moisture potential is reduced.*"

So, this leads me to conclude that attribution of the derived ET responses to VPD entirely as a result of plant physiology – by using a relationship that derive from an acknowledged limitation in its ability to respond to soil moisture – is inaccurate. This attribution is repeated throughout the manuscript and is illustrated in statements like "plants that are evolved to bred to prioritize primary production over water conservation (e.g., crops) exhibit a higher likelihood of atmospheric demand-driven response (found in the abstract)." An alternative interpretation is

that these ecosystem types are responding in this way because they have, on the whole, been subjected to less soil water limitation (due to the non-negligible effects of irrigation?) Without decoupling the effects of VPD from those of soil moisture, I think that the interpretations offered here could be quite misleading.

**References**

Medlyn, B. E., Duursma, R. A., Eamus, D., Ellsworth, D. S., Prentice, I. C., Barton, C. V., ... & Wingate, L. (2011). Reconciling the optimal and empirical approaches to modelling stomatal conductance. *Global Change Biology*, *17*(6), 2134-2144.

Medlyn, B. E., De Kauwe, M. G., Lin, Y. S., Knauer, J., Duursma, R. A., Williams, C. A., ... & Linderson, M. L. (2017). How do leaf and ecosystem measures of water-use efficiency compare? *New Phytologist*, *216*(3), 758-770

MÄKELÄ, A., BERNINGER, F., & HARI, P. (1996). Optimal control of gas exchange during drought: theoretical analysis. *Annals of Botany*, *77*(5), 461-468.

Novick, K. A., Ficklin, D. L., Stoy, P. C., Williams, C. A., Bohrer, G., Oishi, A. C., ... & Scott, R. L. (2016). The increasing importance of atmospheric demand for ecosystem water and carbon fluxes. *Nature Climate Change*, *6*(11), 1023.

Kirschbaum, M. U. (1999). CenW, a forest growth model with linked carbon, energy, nutrient and water cycles. *Ecological Modelling*, *118*(1), 17-59.

Zhou, S., Yu, B., Huang, Y., & Wang, G. (2014). The effect of vapor pressure deficit on water use efficiency at the subdaily time scale. *Geophysical Research Letters*, *41*(14), 5005-5013.

---

## Referee Comment (RC2) · Anonymous Referee #2 · 12 Dec 2018

The authors present a study of how evapotranspiration respond to vapor pressure deficit (VPD) at the ecosystem scale by deriving an analytical equation based on Penman-Monteith equation and an empirical model of stomatal conductance (Eq. 7) and by analyzing flux tower observations in 66 sites of the Fluxnet-2015 dataset. Theoretical and empirical results suggest that ET mostly decreases as VPD increases below a given VPD threshold, this threshold is at quite high VPD values except for crops. The authors attribute this result to a dominant role of plant physiological controls on ET as VPD rises.

The question raised by the authors is important for various fields as ecohydrology, land-atmosphere interactions, biogeoscience and the analysis is interesting and controversial. The postulated concave down relation between ET and VPD is generally

counterintuitive and I think need to be confirmed better with more direct results. This is indeed contrary to our leaf level understanding (and observations) of transpiration response to VPD (see my detailed explanation with reference to literature below, Comment 2). While I am intrigued by the article, I have a number of major and minor comments that hopefully can be solved to support the findings of the study.

(i) Many figures are related to the functional form derived theoretically by the authors (Eq. 7, Eq. 10), however the basic result (e.g., ET vs VPD) is never directly presented in any plot. I would like to see a plot with ET boxplot for different bins of VPDs for different sites (e.g, the partial derivative of ET with respect to VPD mentioned by the authors PP 3, LL 12). This would be important also to understand the uncertainties and potential problems associated with Eq. 10, see for instance my remark on the net-radiation dependence on VPD in the minor comments. Maybe it is obvious and the fit is perfect by construction but it is not very clear to me how Eq. 7 and 10 are fitting the raw data, the Figure 2 with $\sigma$ value is not sufficient to understand this aspect.

(ii) I think the introduction of the article should contain a discussion of what is known about transpiration response at leaf-scale. We know very well that stomatal conductance is reduced in response to VPD (e.g., Oren et al 1999; Damour et al 2010) and many observational studies suggest that transpiration increases with a concave downward response (e.g., Rawson et al 1977; Turner et al 1984; Mott and Peak 2013). Some observations show a reduction of transpiration at high VPD (e.g., Farquhar 1978) the so-called "feedforward response" of stomatal but this behavior was mostly dismissed as an artifact of measurements rather than a true behavior (e.g., Franks et al 1997). Therefore we are left with the fact that transpiration-VPD relation at the leaf scale is mostly positive and surely concave downward. Now there could be several reasons why ecosystem scale response of ET to VPD may be different from leaf-scale, e.g., effects soil moisture limitations, land-atmospheric feedbacks between transpiration and VPD. The presented approach is looking only at one-way response of ET to VPD without accounting for land-atmospheric feedbacks (for instance during dry soil conditions)

P 23 LL 5-6. This has limitations as discussed by the author themselves but also gives too much weight on the plant physiological control. One thing is to attribute the overall result to plant physiological response (e.g., PP 1 LL 8), another is to attribute the observed response to feedbacks and controls acting at the ecosystem scale and not at the leaf-level. Currently, the article is attributing the observed response (concave upward, PP 23, L13) to plant physiology, which is at odd with what we know at the leaf-level. This needs to be clarified in the manuscript and a very convincing explanation needs to be provided, otherwise my current feeling is that other controls are incorrectly attributed to plant physiology. At least for me, it is not "easy" to see the mechanism leading to a change in curvature from leaf-scale to the ecosystem scale.

(iii) In one hand, there is beauty in the newly derived Eq. (7) since should substitute the unknown gs of Penman-Monteith equation with two variables g1 and uWUE, which are theoretically better known (e.g., Table 1, PP 8 LL 26-27). However, this also comes with a risk, because if g1 and uWUE are spatially variable as they could be (especially g1 according to the original publications), we are passing from one unknown to two unknowns. Plus, one variable is representing a plant control (g1) and the other one is somehow representing the response (uWUE), so I am afraid there is a mixing of concepts in the same equation. The author are very confident that their equation captures the relationship between ET, GPP and VPD with the fit of $\sigma^*$ uWUE (P 19, LL 10-12), but the imposed lack of variability of uWUE and g1 and also the assumption on Rn being unaffected by VPD must be mostly trusted. Therefore, I would be more careful in the argumentation and as wrote before I would like to see how ET and VPD are actually related using a binning approach on VPD.

(iv) I think sometime there is an abuse of the "leading-order term/behavior" (PP 16 LL 18, P 18 LL 9, P 18 LL 29), which in mathematical function has a specific meaning but it is unclear how it is used in the context of this article. I would also suggest to separate better the results related to the theoretical derivation (up to Section 3.3 pretty much) from the ones based on empirical data (afterwards)

Minor Comments

PP 1 LL 17. I would not necessary talk of "plant stress". A drop of leaf water potential reduces stomatal conductance but this does not necessarily mean "stress".

PP 1 LL 22-24. Please see also Roderick et al 2014.

P 2. LL 17 . . . could "also" cause a decrease in the likelihood of precipitation

P. 2. LL 33-35. I would tend to disagree, there are many old articles starting from the ones of G. Farquhar (see for instance references in the main comments) that describe humidity (VPD) role on stomatal response. The basic physiological knowledge was established since quite some time. However, I agree that much more uncertainty exists on the ecosystem scale response.

P. 3. LL 2. See also Katul et al 2010

P. 3. LL 9. I would not refer to those as "novel tools".

P. 5 LL 4. It could be worth mentioning that "g1" is the water use efficiency parameter in the "optimal stomatal conductance model".

P.6 LL 4. I guess you are referring to Eq. (4) and not (3) here.

P. 7. LL 14. I would state already at this stage that g1 is derived for each PFT from Medlyn et al 2017 as in Table 2 and uWUE baseline values from Zhou et al 2015.

P. 7. Eq. (10). Please note that in such a derivation the indirect role of VPD in modifying surface temperature and therefore net radiation (Rn) is not accounted for. So technically speaking Eq. (10) is incomplete. This needs to be stated and justified.

P. 10. LL 2 and LL 5. The direct link between ET decreasing with VPD and "physiological controls" is not fully justified as I am discussing in the main comments.

P. 10. Equation (12). There is a "$\sigma$" missing or is intentional? If yes, please explain.

P. 10. LL 11. Also g1 could vary with soil moisture. As a matter of fact, the water use

efficiency parameter of optimality models has been shown to vary with soil moisture (Manzoni et al 2013).

P. 10 Equation (15). Why $\sigma$ is removed from Eq. (13)?

P. 10 LL 29-30. As a matter of fact, WUE is much lower in semi-arid sites than in wet environments because of their lower productivity (e.g., Beer et al 2009). WUE and uWUE are diagnostic variables. It is "g1", which represents a physiological control that should be lower (more water use efficient) in water limited ecosystems and higher where there is plenty of water, as you write in Page 11 LL 7-9. I think the distinction on the role of those two variables (parameters of your model) should be better framed.

P. 11. LL 14-17. uWUE is much more constant than g1 in Table 2, therefore most of these effects should be attributed to variability in g1.

P 13. LL 7-9. In these patterns there could be a significant contribution of water availability with crops that are irrigated and maintain high ET in dry (high VPD conditions) while shrubs are mostly water limited at high VPD levels.

Figure 4. I would suggest modifying the plot and having ga in the x-axis and different temperatures plotted with various lines.

P. 16. LL 16. Maybe I am missing something obvious but observations are also plotted based on the same Equation (10) allowing only variability in $\sigma$ to fit better the data. However, in such a case the functional form is partially prescribed except if $\sigma$ departs significantly. Is there not the risk of some circular reasoning?

P. 16. LL 14. "nearly exactly" is a bit exaggerated, I think.

P 18. LL 32. The issue with low soil-moisture can emerge even when soil-moisture is not extremely low, because of the atmospheric feedbacks that increase VPD.

P.21. LL 1. There are many cases even before getting to extremely dry soil that land-surface could feed back on humidity and VPD (e.g., Rigden and Salvucci 2015) I am

not sure the assumption of constant uWUE and g1 in time and space really works for the majority of the conditions.

P. 21. LL 2-3. I think this is much more relevant than only for extreme conditions and could actually affect the observed behavior much more than currently stated.

P 23 LL 6. Where the 30% number is coming from? I think it is actually quite challenging estimating from observations when soil moisture is limiting evapotranspiration.

P 23. LL 16-21. I think this part should belong more to introduction than results, please see also my major comments on the shape of ET – VPD.

P 24. LL 24. Which exponent are you referring to here?

P 26. LL 2-4. Yes, this is true but at least an overall representation of how ET changes with VPD binning VPD in order to average variability for various conditions should be provided in such an article.

P 27. Appendix A. It would be nice to provide additional information for the Fluxnet sites relevant to this article as VPD, net radiation, temperature, wind speed, and latent heat for the analyzed period during the growing season, number of hours retained for each site for the analysis, etc.

References

Farquhar G.D. (1978) Feedforward responses of stomata to humidity. Australian Journal of Plant Physiology 5, 787–800.

Franks P.J., Cowan I.R. & Farquhar G.D. (1997) The apparent feedforward response of stomata to air vapour pressure deficit: information revealed by different experimental procedures with two rainforest trees. Plant, Cell and Environment 20, 142–145

Turner N.C. Schulze. E,-D, & Gollan T. (1984) The response of stomata and leaf gas exchange to vapour pressure deficits and soil water content. I. Species comparisons at high soil water contents, Oeeologia 63, 338-342,

Rawson HM, Begg JE, Woodward RG (1977) The effect of atmospheric humidity on photosynthesis, transpiration and water use efficiency of leaves of several plant species. Planta 134:5-10

Roderick, M. L., Sun, F., Lim, W. H., and Farquhar, G. D.: A general framework for understanding the response of the water cycle to global warming over land and ocean, Hydrol. Earth Syst. Sci., 18, 1575-1589, https://doi.org/10.5194/hess-18-1575-2014, 2014.

Mott, K. A. and Peak, D. (2013), Testing a vapour phase model of stomatal responses to humidity. Plant, Cell & Environment, 36: 936-944. doi:10.1111/pce.12026

Katul, G.G., Manzoni, S., Palmroth, S., Oren, R., 2010. A stomatal optimization theory to describe the effects of atmospheric $CO_2$ on leaf photosynthesis and transpiration. Ann. Bot. 105 (3), 431–442,

Manzoni, S., G. Vico, S. Palmroth, A. Porporato, and G. Katul (2013), Optimization of stomatal conductance for maximum carbon gain under dynamic soil moisture, Adv. Water Resour., 62, 90-105,

Beer, C., et al. (2009), Temporal and among-site variability of inherent water use efficiency at the ecosystem level, Global Biogeochem. Cycles,23(2), GB2018, doi:10.1029/2008GB003233

Rigden, A. J., & Salvucci, G. D. (2015). Evapotranspiration based on equilibrated relative humidity (ETRHEQ): Evaluation over the continental U.S. Water Resources Research, 51, 2951–2973.

Oren, R., J. S. Sperry, G. G. Katul, D. E. Pataki, B. E. Ewers, N. Phillips, and K. V. R. Schäfer (1999), Survey and synthesis of intra- and interspecific variation in stomatal sensitivity to vapour pressure deficit, Plant Cell Environ., 22(12), 1515–1526.

Damour G, Simonneau T, Cochard H, Urban L. An overview of models of stomatal conductance at the leaf level. Plant Cell Environ 2010, 33:1419–1438. doi:10.1111/j.1365-

3040.2010.02181.x.

---

## Author Comment (AC1) · 18 Dec 2018

Thank you very much for an excellent and very thoughtful review. Addressing your comments will greatly improve the manuscript. We agree with your comments, and think the manuscript could use substantial re-framing and rewording to clarify how we are answering our research question, and how the answer may vary with climate and environmental factors.

Just a quick general comment before addressing your specific comments. By including a PFT-focused analysis we did not fully communicate the major goal and scope of our project: we are trying to characterize the response of ET to VPD, with all other environment variables held fixed. To accomplish this, we need to formulate an explicit function of ET in terms of environmental variables and parameters, where any parameters can be approximated as constant with regards to some [arbitrary] VPD perturbation scenario. We will elaborate more below on why we used PFT-focused scenarios for much of our analysis. However, the goal of our manuscript was much more simple and fundamental: at a given place or time, if you introduce a perturbation to VPD, what is the immediate ET response (e.g. positive or negative)? Answering this question does not require the much stronger assertion that any parameters must be invariant within a given PFT.

**1 Compounding uncertainties**

We agree that the uncertainties are large both within a PFT and across PFTs. Looking back, we believe that some of our language communicating *Lin et al.* (2018), *Medlyn et al.* (2017), and *Zhou et al.* (2015)'s results was misleading, and this was exacerbated by our focus on PFT-analysis. We will remove that language from the manuscript, and add language to better communicate our results as reflections of the considerable within-PFT uncertainty.

We think comparing the rankings of given PFT values for $g_1$ and uWUE can be misleading, given the magnitude of the uncertainties involved. Most of our calculated values for uWUE are within one standard deviation of *Zhou et al.* (2015)'s results, but these deviations can result in some changes in the ordering of PFTs from high to low uWUE. The bigger problem in our eyes is that we made a mistake with some language suggesting within intra-PFT variability is less than inter-PFT variability, but clearly this is not the case. We will remove this language. This misinterpretation should not have been in the manuscript and actually contradicts other manuscript content; for example, we included *Zhou et al.* (2015)'s results on uWUE in Table 2

explicitly to be transparent about the within-PFT uncertainties. $g_1$ values also exhibit considerable uncertainty, and again, while the relative rankings of these values may change, they are all within observed ranges in *Medlyn et al.* (2017). We did not include these in Table 2 because *Medlyn et al.* (2017) does not provide the numerical values; however, we think it would be useful to provide estimates of these ranges from *Medlyn et al.* (2017)'s figures in our Table 2.

Regarding how to interpret $g_1$ and uWUE: we think the best way is with established physical relations from *Medlyn et al.* (2011) (Equation 3 in the manuscript) and *Zhou et al.* (2014) (Pg. 7, line 10 in the manuscript), with the knowledge that there is uncertainty involved. The quantities in these relationships all have some intrinsic physical meaning, but can vary substantially within a PFT. Again, we need to alter our language to better reflect this intra-PFT variability.

However, *Zhou et al.* (2014) did establish a constant uWUE approximation as a good approximation for capturing the relationship between GPP, ET and VPD at a given place and time, so it is still a very useful and robust approximation for answering our research question (see discussion in the introduction of this comment). Additionally, a constant $g_1$ approximation is used in many earth system models, so while it introduces uncertainty, it also makes our framework useful for interpreting modeled vegetation response in ESMs and GCMs.

We also agree with the comment that the best way to think of this is probabilistically. This is the most robust approach to dealing with approximations - we introduce randomness to our variables and parameters to account for all of the physics that are not explicitly accounted for, as well as observational uncertainty. However, as far as we know we still have not developed a general, robust, and efficient arithmetic for random variables. We could try and adapt a Bayesian model to this problem, but given the large amount of arithmetic involved, fully incorporating a Bayesian representation to every variable in the analysis would be a very hard problem, and a significant research project in its own right. We think a good compromise is to add language directing the reader to interpret our results more probabilistically, which we have already presented probabilistically in the figures. For example, in Figure 5 the range of values in each plot represents a range of possible responses in the sign term. Within this figure, it's worth noting that the intra-PFT variability is greater than the inter-PFT variability, which is consistent with some of the previous results you highlight. Again, we need

to add language highlighting this, and its consistency with previous results. We included this variability and uncertainty explicitly to be transparent. A more difficult question is how much of this variability is due to observational and model error, and how much of it is due to climate and plant physiological variability (see Section 3.5).

**2    Attribution to physiological responses**

We do not want our analysis to be interpreted as an assertion that $g_1$ and uWUE are attributes of PFT only. We expect them to vary both within a PFT and across a PFT. Our primary goal in using $g_1$ and uWUE was to develop an explicit expression of ET as a function of environmental variables, and use this to assess the response to a change in VPD. We focused our analysis using PFTs because this is how it was framed in previous studies in the field, and specifically the studies used in our derivation (*Zhou et al.*, 2014, 2015; *Medlyn et al.*, 2017). We were originally thinking the PFT-focused analysis could be useful, especially given that climate models generally hold plant parameters fixed with respect to PFT, so long as we were transparent about the large uncertainties and problems with this approach (see Figure 5).

We agree with you that stating that any quantity is fixed within a PFT is hard to believe. Phenotypic variation and adaptation within a given *species* can be considerable (i.e. effecting $\lambda$, $g_1$, and uWUE), so it would be hard to say that anything would be constant within a PFT made up of many different diverse species. Both phenotypic variation as well as the species distribution and dominant PFT at a given location will all be strongly optimized in response to climate. In this sense, the distribution and evolution of local climate is a strong control on the local structure and physiology of a given ecosystem.

What we need to communicate better is that a given ecosystem's state at a place or time controls its response to a VPD perturbation. We are making an approximation that we can parameterize the effect of the ecosystem's state on VPD response with $g_1$ and uWUE. These quantities can also vary due to soil moisture condition; the approximation we need to answer our research question is that they are fixed with respect to a VPD perturbation. In this sense, we do not view the two statements: "plants that are evolved to bred to prioritize primary production over water conservation (e.g., crops) exhibit

a higher likelihood of atmospheric demand-driven response" and "ecosystem types are responding in this way because they have, on the whole, been subjected to less soil water limitation (due to the non-negligible effects of irrigation?)" as mutually exclusive. In fact, we view them as consistent with a view that crops and their physiology (parameterized by $g_1$ and uWUE) exist at a given time and place because they have not been subjected to soil water limitation, and they have a given response to VPD because of their physiology (which is a direct effect of the environment). In this way, climate and land surface state are causes of the VPD response both directly and through their effect on plant physiology (parameterized by $g_1$ and uWUE).

When writing this manuscript, there was definitely some internal tension and debate about how to best frame the analysis and results. We could either focus on PFT-oriented results as previous literature has done (e.g. holding plant physiology fixed within a given PFT), or allow plant physiology terms to vary through time and space and look at the distribution of ET response to VPD (see Section 3.5). After re-examining the manuscript both after some time away from the problem and in light of your comments we think a strong argument could be made that we made the wrong choice with respect to this focus. It may have made more sense to focus our analysis on the ET response more generally across space and time as ecosystem-scale plant physiology varies in response to climate and soil moisture.

**3   Next Steps**

In order to improve the manuscript and our communication of the answer to the question "When does VPD drive or reduce ET?", we see a few potential paths that we will consider between now and the final response after the discussion period. If you (or anyone else) has any opinions or comments in the meantime, we would appreciate the insight and feedback.

- **Option 1:** We include the discussion presented in this review and response, and include language explicitly stating that the purpose of the PFT analysis is to provide connections to other PFT-constant analyses and models (e.g. ESMs and GCMs). We will add extensive language on the uncertainty and weaknesses of this approach, and reframe our conclusions to reflect his uncertainty.

- **Option 2:** We instead alter our analysis to look at how VPD response

varies with climate and general plant physiological variability, instead of focusing only on PFT analysis that poorly captures all of the observed variability in ET response. Sections 3.1 - 3.2 would be replaced by analysis directly relating ET response to general climate and plant physiological terms, rather than PFT-mean analysis. For example, the "scaling term" analysis would be presented in terms of generic changes in plant height and temperature, and the "sign term" analysis would be framed by generic changes to $g_1$ and uWUE, as informed by previous literature.

- **Option 3:** This is the most extreme option in terms of modifying the manuscript. We significantly alter the manuscript and instead focus on just the general shape of the ET-VPD curve (with environmental variables held fixed). This was not discussed in the review, but one of our most noteworthy results is on the general shape of the ET-VPD curve being concave up, given an assumption of a square-root VPD dependence of ET. This result is independent of any assumptions of uWUE and $g_1$, and to our knowledge it is first derived curve of ecosystem response to VPD (see Section 3.7). It also highlights the importance of discerning the exact exponent of VPD dependence, as this alters the fundamental nature and shape of the curve. Essentially, the manuscript would become just our motivation and derivation, followed by Section 3.7 exploring the consequences of the derivation for the shape of the VPD curve. This alteration of the manuscript would likely result in more of a technical note-type paper, and we are hesitant to do this because we think there is still a lot of useful information obtained by tying our results to real-world scenarios (as in Options 1 and 2 above).

Here we present some final minor comments and concerns on the technical details of our proposed changes to the manuscript, which are relevant to representing uncertainty and spatiotemporal variability in uWUE *and* $g_1$. In the original manuscript we used a single $\sigma$ term to represent this variability. We did this because changes in uWUE and $g_1$ induce a very similar change in the ET solution, which made solving for independent $\sigma_{uWUE}$ and $\sigma_{g1}$ terms intractable at a given time and place, and representing variability in both $g_1$ and uWUE difficult. We decided to hold $g_1$ fixed within a PFT for a two reasons: 1) ESMs and GCMs generally hold $g_1$ fixed, and 2) letting uWUE vary seemed more appropriate to represent specifically soil water variations

on stomatal conductance, as uWUE modifies stomatal conductance analogously to how soil moisture factors modify a maximum stomatal conductance in land surface models (e.g. it is a multiplicative factor on the entire stomatal conductance term). Because uWUE and $g_1$ induce similar changes in ET, at least qualitatively we think that having a single $\sigma$ can represent some of the variability in both $uWUE$ and $g_1$. However, formally there is no explicit variability in the $g_1$ term. The discussion in this review rightly points out that based on previous results we do expect some variability in $g_1$ as well. So, an unresolved question is how important is it to the analysis and its interpretation to include an explicit $g_1$ variability term in addition to the existing $\sigma$ term. We are including these comments to hopefully stimulate some discussion on the importance of explicitly representing $g_1$ variability, given the difficulties of doing so within our framework. However, we could imagine some timescale based approaches where we might be able to account for both the $g_1$ and uWUE variability, for example by making assumptions over what time scale each quantity is fixed, and fitting based on that (e.g. $g_1$ is fixed for a given season, and uWUE is fixed for a given day). This approach might allow a tractable solution, and also could help us filter model error and observational noise from "true" plant physiological and climatic variability in uWUE and $g_1$. The cost of all of this is a significant increase in the analysis and content of the paper, as well as some increased opacity to the methods. Comments are welcome.

Thank you again for the thoughtful review. We hope it stimulates further discussion.

**References**

Lin, C., P. Gentine, Y. Huang, K. Guan, H. Kimm, and S. Zhou (2018), Diel ecosystem conductance response to vapor pressure deficit is suboptimal and independent of soil moisture, *Agricultural and Forest Meteorology, 250,* 24–34.

Medlyn, B. E., R. A. Duursma, D. Eamus, D. S. Ellsworth, I. C. Prentice, C. V. M. Barton, K. Y. Crous, P. D. Angelis, M. Freeman, and L. Wingate (2011), Reconciling the optimal and empirical approaches to modelling stomatal conductance, *Global Change Biology, 17*(6), 2134–2144, doi:10.1111/j.1365-2486.2010.02375.x.

Medlyn, B. E., M. G. De Kauwe, Y.-S. Lin, J. Knauer, R. A. Duursma, C. A. Williams, A. Arneth, R. Clement, P. Isaac, J.-M. Limousin, et al. (2017), How do leaf and ecosystem measures of water-use efficiency compare?, *New Phytologist*, *216*(3), 758–770.

Zhou, S., B. Yu, Y. Huang, and G. Wang (2014), The effect of vapor pressure deficit on water use efficiency at the subdaily time scale, *Geophysical Research Letters*, *41*(14), 5005–5013.

Zhou, S., B. Yu, Y. Huang, and G. Wang (2015), Daily underlying water use efficiency for AmeriFlux sites, *Journal of Geophysical Research: Biogeosciences*, *120*(5), 887–902, doi:10.1002/2015jg002947.

---

## Referee Comment (RC3) · Anonymous Referee #3 · 20 Dec 2018

The manuscript by Massmann, Gentine and Lin explores how (evapo)transpiration rate responds to changes in vapor pressure deficit (VPD), employing a combination of model developments and data. The topic is interesting and important, with implications for different disciplines. Nevertheless, I have some fundamental concerns regarding the theoretical developments presented in the manuscript and how to best present the results.

Methodology:

Even though not reported in the manuscript, the optimization model parameter $g_1$ is a function of the marginal water use efficiency $\lambda$ (see e.g. Medlyn 2011 Global Change Biology). Neglecting such functional dependence has at least two main consequences regarding the theoretical developments and hence the results.

[Figure]

1) The authors justify neglecting the role of soil moisture by stating that uWUE has been shown to be relatively constant. I disagree with this conclusion. $\lambda$ is known to be changing with soil water availability (Manzoni et al 2011 Functional Ecology; Zhou et al 2014 Agricultural and Forest Meteorology). As such, I consider the results discussed in the manuscript problematic, not only at extremely low soil water contents, also in the light of complex (and differing) distributions of soil moisture in the different PFTs (at least two sites exhibit a bimodal distribution; Figure 7). I thus wonder if it would be more robust to consider the soil moisture variability in the theoretical development, albeit in a simplified manner. This would greatly enhance the impact of this work and usability of the theoretical developments, because there are no doubts that soil water availability affects ET rate and, as such, may mediate its response to VPD. Furthermore, it may even reduce the 'uncertainty' $\sigma$, by actually accounting for the mechanisms at play. A simpler (but less powerful) approach would be to restrict all (or most of) the analyses and discussions to well-watered conditions, but this would result in number of observations being reduced and changing significantly from ecosystem to ecosystem.

2) It is assumed that the uncertainty parameter $\sigma$ modulates uWUE, but not $g_1$ (or any other parameter). I understand the idea of focusing the uncertainty analysis on the most uncertain parameter, but, given the strong relation between uWUE and $g_1$, I question this approach. Moreover, given that both uWUE and $g_1$ are affected by soil water availability, considering the uncertainty of one of the two only, is problematic. This undermines the conclusion that any systematic bias stemming from neglecting the effects of the soil water content should be 'absorbed' by $\sigma$ and, implicitly, that the proposed method does not work only in the cases where a relation between $\sigma$ and soil water content emerges.

More in general, the method section would benefit from a number of clarifications.

- Several of the symbols are not defined at their first appearance and/or do not appear in the table (e.g., $R_{net}$, $g_0$); some symbols have two meanings (e.g., $T$, which is used both as temperature and transpiration rate); and some others have the same meaning

despite being different (e.g., $R_{air}$ and $R$). Particular care should be posed in defining $\lambda$, because this symbol has been used to denote both $\frac{\partial E}{\partial A}$ and $\frac{\partial A}{\partial E}$, depending on the publication.

- As noted above, the dependence of $g_1$ on $\lambda$ should be clarified, or else a reader not familiar with Medlyn et al (2011) work is left wondering what the relation between $g_1$ and uWUE is (in its present form, the manuscript just hints at a possible relation – P7, L7; but the relation is strong, as both are functions of $\lambda$ and $\Gamma$).

- It remained unclear to me why the authors chose to present also the leaf level stomatal conductance, when, in the end, they then use a canopy scale stomatal conductance. The latter is derived from the leaf level stomatal conductance, but such derivation has been published elsewhere.

- A range of observed $g_a$ are considered (e.g., in Fig. 4). I think it is worth mentioning how the aerodynamic conductance was determined. This affects the validity of the conclusion that wind conditions play a secondary role (P14, L20).

- The FluxNet data are used to determine all the terms in Eq. 7, directly or via fitting, as stated on P7; L13. Later on, apparently the same data are used to determine $\sigma (Eq. 9)$. I think it would be best to clarify how these two uses of data 'co-exist'. I suppose uWUE is first determined, assuming $\sigma = 1$; and then, for the obtained uWUE, $\sigma$ is determined.

Presentation of results:

The authors discuss in detail the sign and value of $\frac{\partial ET}{\partial VPD}$. This makes sense, given the aim of the work, but I think that the distinction in scaling term and sign term is more confusing than clarifying: the 'sign' term affects also the magnitude of $\frac{\partial ET}{\partial VPD}$. I wonder if it would be cleaner to really focus on the sign of the derivative (i.e., Eq. 13 and the bottom part of Figure 3) and then discuss the overall magnitude of the derivative, without distinguishing between the two terms. This would also reduce the number and complexity of figures. Alternatively, the authors could try to interpret the terms in $\frac{\partial ET}{\partial VPD}$

as largely 'plant driven' and 'environment driven'. This however poses the question of to which extent $g_a$ is determined by the environmental conditions (chiefly wind speed) vs. plant/canopy features (chiefly, canopy height).

More in general, the results presented in this manuscript are many and it is not easy to see the logical connection among the different aspects discussed. The take home messages would emerge more clearly, should the presentation of results be stream-lined. Thus, steps should be taken to simplify the presentation of results. For example, Table 4 presents the 'bulk statistics' of the observed $\frac{\partial ET}{\partial VPD}$ and their match with the theory and then Figure 5 somehow re-iterate the conclusion, but now breaking down the data. Also, Section 3.7 comes a bit as a surprise and, in a certain way, it would fit better in the Methods.

Finally, some of the conclusions are not fully and quantitatively supported. Examples are P16, L20 where reference is made to 'a bit more variability' (no quantification of 'a bit'), or the use of the expression 'leading order behavior'.

Minor comments

Introduction: The introduction would benefit from a more thorough review of what is known about transpiration response to VPD. Examples are Oren et al 1999 (Plant Cell and Environment) and, more recently and based on FluxNet data, Novick et al. 2016 (Nature Climate Change), also discussing the effects of soil water content. Further-more, the discussion on plant strategies reported here feels a bit oversimplified. While clearly it is beyond the scope of this work to discuss plant-plant interactions and com-petition for water or other strategies like CAM photosynthesis, the current text implicitly suggests these other factors do not exist. For example, the 'ultimate' adaptation to exploit times of low VPD is CAM photosynthesis, where stomata are (mostly) opened during the night. Also, lack of soil moisture conservation not being a sensible strat-egy is most likely correct when considering a uniform stand (or isolated vegetation), but competition for resources may affect what a sensible strategy is. In this respect,

one additional feature of crops is that they are (generally) planted in an even aged monoculture, while other ecosystems may be characterized by mixed species/ages.

P 9: The first part of the Results and discussion appears to belong to the Methods, entirely or at least Eq. 11 and 12.

P13, L32: the effect of temperature is also direct, not just through Clausius-Clapeyron.

P16, L15: 'between' should be removed (or the sentence revised)

---

## Author Comment (AC2) · 23 Dec 2018

Thank you for a very thoughtful review; as in our response to RC1, we feel very lucky to have received such quality reviews.

We read through your major and minor comments, and do not anticipate issues reconciling your minor comments for the final response after the discussion period. We will focus our response on your major comments, with the goal of opening the door to further discussion.

Also please note our response to RC1, as some of the discussion there is relevant to this response; particularly discussion on PFT-scale analysis and variability of $g_1$ and uWUE (RC1 Response).

**1 Major comment (i)**

One issue with analyses attempting to diagnose the ET response to VPD with pure data (e.g. using binned boxplots, as you suggest), is that ET varies strongly with both aerodynamic conductance and radiation. Simply binning by VPD or by site includes too many confounding factors to really diagnose what the attributable response is to VPD. This issue is really what motivated our analysis and makes it unique; by building an analytical framework based on confirmed results in the literature (uWUE, Medlyn model, Penman-Monteith [PM]), we are able to formally deduce what the actual ET response is, with other environmental factors held fixed. This "other environmental factors held fixed" component is, in our mind, nearly impossible to deduce from pure data, because the environment is always changing and direct analogues are either impossible to find or result in too small of a sample size to form a meaningful conclusion. However, while we cannot assess our formulation of the derivative with respect to VPD, we can assess the actual model of ET, which includes all approximations we introduced (specifically uWUE). As a part of our analysis test suite we generated a figure of this comparison and reproduce it here (Figure 1). Apologies this is not a publication quality figure, but we wanted to get this response out sooner rather than later to stimulate discussion, especially as we are already behind because of the AGU Fall Meeting. This comparison is fair in the sense that by using uWUE we introduce an extra free parameter, so in the original PM model we introduce an extra free parameter modifying $g_s$ and fit it also for each given PFT. The introduction of the uWUE approximation does degrade the quality of the ET model relative to the original PM model. However, we expect this, as using uWUE is a simplification over using GPP directly,

[Figure]

Figure 1: Mean bias (absolute value) and root mean squared error (RMSE) for three different ET estimates, relative to FLUXNET-2015 observations: original (original Penman-Monteith formulation, e.g. Equation 1 in the manuscript, with Equation 4 providing $g_s$); uWUE (Equation 8 in the manuscript); iWUE (as in Equation 8, but if we had used iWUE to remove GPP dependence rather than uWUE).

and may break down in extreme, limiting cases like when VPD, ET or GPP go to 0. Additionally, we expect errors introduced by assuming that uWUE is fixed within a given PFT. These errors may be reduced if we account for spatiotemporal variation in uWUE, as discussed further in our response to RC1. As a purpose-built model for assessing the ET response, we think the uWUE approach is justified. Essentially we are saying that to the degree we trust arithmetic, uWUE, Penman-Monteith, and the Medlyn stomatal conductance model, our evaluation of the derivative is robust.

However, the discussion so far assumes that the variables in our model (Equation 8) do not change directly with VPD and we are free to evaluate the derivative. You bring up a very good example that we did not consider: a change in VPD will induce a change in net radiation through surface temperature that PM does not consider. We will definitely add language addressing this oversight. Ideally, we would incorporate that effect directly

in Equation 8. We briefly tried to do this in time for this response, but difficulties deriving a diagnostic equation for temperature sabotaged the effort, and we do not want to delay the response further. We will continue to work on this for the final response phase. In the meantime, we think we can make some logic-based arguments in the interest of discussion as to what the effect would be. In the case where the aerodynamic and physiological ET response (which the manuscript does consider) increases with VPD, we would expect a decrease in surface temperature. This would induce an increase in net radiation, as OLR and ground heat flux decrease. This increase in net radiation would induce a further increase in ET response. Granted, the net radiation perturbation would damp the negative temperature response to the aerodynamic/physiological terms, but all of the signs of the change would stay the same. The converse is true for a decrease in ET with respect to VPD: surface temperature increases, OLR and ground heat flux increase, net radiation decreases, ET decreases. In this way, we believe the effects of VPD on radiation do not change the nature of the sign of the response, but could amplify the magnitude. Comments on this are encouraged, and we will work to include a more quantitative and rigorous analysis in the final response. Sometimes it is easy to trick oneself with these types of qualitative arguments, particularly with complicated systems, so hopefully we have not presented faulty logic in our rush to make this response public.

**2 Major comment (ii)**

Thank you for having an open mind about the concave up ET-VPD curve result and providing clear explanations of concerns about this result based on literature. Given the apparent controversy, we will definitely add more language connecting previous results, particularly at the leaf scale, with the presented results. Here we will informally elaborate in the interest of discussion.

One of the primary differences between ecosystem scale assessments of ET (specifically using Penman Monteith) and experimental and modeling results of leaf scale response to VPD is that by taking an energy balance approach Penman Monteith consistently accounts for the effects of the energy cost of evaporating water from a surface (e.g., changes in $e_{sat}(T)$). To demonstrate this, compare Figure 2 and Figure 3 ("Original Penman Monteith" Curve). In Figure 2 we apply a conceptual leaf scale approach to calculate ecosystem ET without the energy balance consideration introduced by PM. In this figure, the curve shows a concave downward shape, in direct analogy to leaf scale T. Once the effects of evaporation's thermodynamics are included in the energy balance, the curve's shape is no longer concave downward (Figure 3, "Original Penman Monteith" curve). How these conceptual results translate to leaf scale experiments depends on the experimental setup (e.g., *Rawson et al.*, 1977; *Turner et al.*, 1984; *Mott and Peak*, 2013); however generally chamber-based leaf scale experiments do not preserve the energy balance relationships we expect for a surface in a natural environment (sometimes intentionally by design). Additionally, we would like to note that a concave up result is not necessarily inconsistent with the statement "transpiration at the leaf scale is mostly positive" with VPD. Especially for plants in well watered environments like crops, the manuscript's ecosystem scale shape (Figure 3) would not be out of place on the figures in *Rawson et al.* (1977) and *Mott* (2007) for the range of VPD considered (Figure 3 compared to Fig. 2 in *Rawson et al.* (1977), also Figure 7 in *Mott* (2007)). However, the change in relative magnitudes are not similar, and this effect can be traced partially to the effect of decreasing surface $e_{sat}(T)$ in response to increasing evapotranspiration. While not directly considered here (or in the manuscript) we also think it is important to consider how plant physiologic response in the natural environment could vary from plants grown in well-watered conditions in experiments. It is not inconceivable that phenotypic variability and adaptation in response to water limitation could result in plants with different responses to those grown in a lab. We presented crops in our figures because they are the most likely to well watered and analogous with laboratory results; however for other PFTs the concavity of the VPD-ET curves is more pronounced at higher VPD as compared to Figure 3.

The previous paragraph discussed the change in frame of reference induced by using PM (and energy balance) to assess ecosystem scale ET rather than directly applying leaf-scale logic from controlled experiments. However, our uWUE-based PM framework additionally accounts for changes in photosynthesis induced by VPD (Figure 3). Discussion in *Damour et al.* (2010) ("The issue of co-regulation of $g_s$ and $A_{net}$") provides some interesting background both on the general weaknesses of current stomatal theory, and specifically on the possible importance of resolving the effects of water stress on $A_{net}$ in stomatal conductance. Given that the current state of the art in physically-derived theory has not converged on the proper way to account for changes in $A_{net}$ and $g_s$ (*Damour et al.*, 2010), we believe using the validated

[Figure]

Figure 2: The conceptual relationship between ET with VPD using a model analogous to leaf scale models and experiments, evaluated with the study's median ecosystem scale resistances for the crop PFT. This estimation of ET is by definition not physically representative, but the figure is intended as a conceptual description of how leaf-scale theory's ET-VPD relationship deviates from energy balance ecosystem theory (Fig. 3). This figure is demonstrative, and not intended to be of publication quality.

[Figure]

Figure 3: The conceptual relationship between ET and VPD using the uWUE derived version of Penman Monteith (Equation 8 in the manuscript) and the original Penman Monteith formulation (Equation 1 in the manuscript), evaluated at the study's median conditions for the crop PFT. uWUE Penman Monteith includes the effects of VPD on photosynthesis. This figure is demonstrative, and not intended to be of publication quality.

uWUE empirical results (e.g. Figure 3, *Zhou et al.*, 2014) is a good approach to develop an estimate of ET response to VPD accounting for changes in photosynthesis. This method of using an empirical result in the absence of a complete physical theory has been used often in science (e.g. turbulence) to increase our knowledge about a given system's behavior, and we think it is well motivated here. However, it is important to add language on how introducing this empiricism could alter interpretation of our results. In particular, while uWUE was developed using leaf scale photosynthesis theory, it was validated with observations of a fully coupled land-atmosphere system. It captures a relatively consistent observed relationship between GPP, ET, and VPD as they vary. There is a chance that some of this relationship could be due to feedbacks between the land and the atmosphere, even though the relationship was derived by leaf scale theory without feedbacks (e.g., if the assumptions behind the uWUE theory introduced "real" errors that were compensated for by feedbacks or ecosystem-scale processes in the observations). Were this the case, then we introduced a conceptual impurity to the statement "we are examining the one way response of ET to VPD." But, given the theoretical motivations for uWUE we think the portion of the relationship related to land-atmosphere feedbacks are likely small, and regardless we argue that this approach is still an improvement over ignoring the documented effect of water stress on net photosynthesis.

Given the continued gaps in our ability to represent stomatal behavior (*Damour et al.*, 2010), particularly for ecosystems in natural environments, we think it is hard to rule out our results. In the end, it comes down to the degree to which we trust our three tools: the Medlyn model, Penman-Monteith, and uWUE. To us these offered the most compete analytical approach to the problem at present, and are supported in the literature. However, it's important to emphasize that we are not saying this is certainly correct, and to do so would be irresponsible given the continued development of stomatal theory and how it translates to the ecosystem scale. What is important is to recognize that something as fundamental as the shape of the ET-VPD curve is still very much an area of open research, and sensitive to the representation of plant physiology (of which there are many possibilities) in a given model or framework. We thank the reviewer for having an open mind and giving us a chance to elaborate; others in the community have rejected this result off-hand, which reinforces for us the importance of demonstrating current uncertainty in VPD-ET relationships.

Just a quick final note:

We think that we miscommunicated with our (mis)use of "plant physiology" throughout the manuscript, and will alter our language in the final response phase to be more precise. Many times when we refer to "plant physiology" we mean any effect (direct or indirect) of an ecosystem response, e.g. any behavior that would not be observed over a wet or bare soil surface.

**3 Major comment (iii)**

Yes, through our manipulation of PM we have replaced stomatal conductance with two parameters ($g_1$ and uWUE). However, given that any physically reasonable representation of stomatal conductance will include more parameters (usually at least some sort of VPD-related parameter [$g_1$], and a model for photosynthesis with more parameters), we think we are actually reducing the number of unknowns in our model for ET. We agree with you that in reality $g_1$ and uWUE are spatially variable, and we elaborate in detail in our response to Reviewer 1 (RC1 Response) on why and how it might be better to frame our results more in terms of this variability. Also, see our comments in Section 1 on the effect of VPD on $R_n$ and the suitability of a binning approach to VPD.

We also agree with you that uWUE includes a representation of ecosystem response. However, we would argue that this response dependence of stomatal conductance is not introduced by uWUE, but is actually an issue that exists inherently in any stomatal conductance theory that includes a photosynthesis term: photosynthesis depends on the physiological response and environmental conditions. Using uWUE is our (approximate) solution to this problem, and it works because we are able to algebraically manipulate PM to isolate ET after introducing uWUE. We do recognize that using uWUE introduces some uncertainty, but as previously discussed in this response we believe it is a useful tool for representing the photosynthesis dependence of stomatal conductance in an analytical framework, with the specific goal of examining VPD sensitivity.

**4   Major comment (iv)**

We agree on both counts: use of "leading order" is misguided and confusing in this context. Also, better separation of theory and empirical results would help communication/understanding.

Thank you again for such a thoughtful review. Including these insights will really improve the manuscript in the final response phase. In our response to Reviewer 1 (RC1 Response) we also laid out some alternatives to our approach for the final review phase not explicitly discussed here. If you (or anyone else) has any opinions on that please weigh in.

**References**

Damour, G., T. Simonneau, H. Cochard, and L. Urban (2010), An overview of models of stomatal conductance at the leaf level, *Plant, Cell & Environment*, *33*(9), 1419–1438, doi:10.1111/j.1365-3040.2010.02181.x.

Mott, K. A. (2007), Leaf hydraulic conductivity and stomatal responses to humidity in amphistomatous leaves, *Plant, Cell & Environment*, *30*(11), 1444–1449, doi:10.1111/j.1365-3040.2007.01720.x.

Mott, K. A., and D. Peak (2013), Testing a vapour-phase model of stomatal responses to humidity, *Plant, Cell & Environment*, *36*(5), 936–944, doi:10.1111/pce.12026.

Rawson, H. M., J. E. Begg, and R. G. Woodward (1977), The effect of atmospheric humidity on photosynthesis, transpiration and water use efficiency of leaves of several plant species, *Planta*, *134*(1), 5–10, doi:10.1007/BF00390086.

Turner, N. C., E.-D. Schulze, and T. Gollan (1984), The responses of stomata and leaf gas exchange to vapour pressure deficits and soil water content, *Oecologia*, *63*(3), 338–342, doi:10.1007/BF00390662.

Zhou, S., B. Yu, Y. Huang, and G. Wang (2014), The effect of vapor pressure deficit on water use efficiency at the subdaily time scale, *Geophysical Research Letters*, *41*(14), 5005–5013.

---

## Author Comment (AC3) · 23 Dec 2018

Thank you for a very thoughtful review. This third review adds further to our feeling that we are extremely lucky to have received such quality reviews.

We agree with your comments and will respond with some general views to open the door for further discussion in this discussion phase. We look forward to reconciling all reviews with a formal response in the final response phase. Also of relevance to this review is our response to RC1, which we will refer to as needed to avoid repeating content.

[Figure]

**1   Soil moisture, uWUE and $g_1$**

Environmental conditions, particularly soil moisture, will effect both uWUE and $g_1$ as you point out. Not accounting for this variability by using a PFT-focused analysis is problematic. We discuss this in our response to RC1, specifically Sections 2 and 3, and think a strong argument could be made that we should have framed our analysis in terms of more general uWUE and $g_1$ variability instead of focusing primarily on PFT variability. If we let uWUE and $g_1$ vary in space and time, then we include a measure of soil moisture through its effect on uWUE and $g_1$. In this case, interpretation of our results relies on a less restrictive approximation that soil moisture conditions remain relatively fixed with respect to the VPD perturbation, which is consistent with the general interpretation of our results (e.g. we evaluate the ET response to VPD with other environmental conditions held fixed).

There are also issues with only including uncertainty in uWUE. We discuss reasons for this approach in our response to RC1 (specifically the final paragraph), and we think there may be some tractable approaches to representing uncertainty and variability in both uWUE and $g_1$ for the final manuscript.

**2   Methods/Introduction**

A more thorough discussion in the introduction about what is known about the transpiration response to VPD is definitely needed. This is also discussed peripherally in our response to RC2. The clarifications you suggest for the methods section will also improve the manuscript.

[Figure]

**3 Results**

Regarding the presentation of the results, we agree that in its current form the manuscript can be disorienting. Just discussing the sign of the derivative, and then discussing the magnitude (while including what the current manuscript refers to as the "sign term") seems to be a much cleaner approach, possibly also differentiating between environmental controls versus plant/canopy controls.

In our response to RC1 we discussed a few revision approaches regarding the results, some of them including significant changes, which should help both with acknowledging $g_1$ and uWUE variability and streamlining the results. One example ("Option 2") is to remove the PFT focus from the first portion of the results (Section 3.1-3.2; this could also go in Methods as you suggest), and instead focus generally on how changes in the environment (e.g. wind speed, temperature) and land surface terms (e.g. $g_1$, uWUE, canopy height) influence the ET response. By using a more sophisticated representation of uncertainty we could then present the distribution of observation-informed ET responses as a function of the model terms ($g_a$, uWUE, $g_1$, etc.), and finally tie all this to either PFTs or specific sites with maps of statistics from the distribution of model parameters and ET response. While this could add a lot of analysis and content to the paper, we think in the end it could streamline the results and their interpretation. Basically we would start with the idealized ET response as a function of parameters (no uncertainty), to the observed ET response as a function of parameters (with uncertainty), to how both the parameters and ET response (with uncertainty) map to specific sites, years and/or PFTs. Thoughts on this are welcome.

We also acknowledge that we need to be more precise about some of our language (e.g. "leading order behavior", "bit more variability"), and thank you for the other minor comments as well; fixing these will improve the manuscript.
553, 2018.